# Comparing Economics, Environmental Pollution and Health Efficiency in China

**DOI:** 10.3390/ijerph16234827

**Published:** 2019-12-01

**Authors:** Zhen Shi, Fengping Wu, Huinan Huang, Xinrui Sun, Lina Zhang

**Affiliations:** 1Business School, Hohai University, Nanjing 211100, China; 20051726@hhu.edu.cn; 2School of Business Administration, Hohai University, Changzhou 213022, China; 1763510216@hhu.edu.cn (H.H.); 1763510127@hhu.edu.cn (X.S.); 20191001@hhu.edu.cn (L.Z.)

**Keywords:** two-stage dynamic DEA model, pollutant emissions, production efficiency, health efficiency

## Abstract

As the modern economy develops rapidly, environmental pollution and human health have also been threatened. In recent years, relevant research has focused on subjects such as energy and economic, environmental pollution and health issues. Yet this has not considered the use of water resources and the impact of wastewater pollutant emissions on the economy and health. This article has combined the following factors like water consumption with wastewater discharge, pollutant concentration in sewage and local medical care expenditure and put them into the model of water resources, energy and health measurement, and a two-stage dynamic data envelopment analysis (DEA) model considering undesirable outputs is applied to 30 provinces (including autonomous regions and municipalities) to calculate the total efficiency, production efficiency and health efficiency in 2014–2017.The results show that the total efficiency values of most provinces are between 0.2 and 0.4, providing large room for improvement. Production efficiency and health efficiency have increased in recent years, but the health efficiency values of most provinces are still so low that they have dragged back the overall efficiency. The key impact indicators of different provinces are different, and each province should formulate different policies according to its own specific conditions so as to purposefully to deepen the energy, economic and medical reforms in each province, and also to promote sustainable economic development while improving health efficiency.

## 1. Introduction

Since the industrial revolution, while the economy has developed rapidly, water and energy on Earth have also been rapidly consumed. According to statistics, by the end of 2018, China’s total water resources reached 279 billion cubic meters, accounting for 6% of the world’s total water resources, ranking fourth in the world, but the per capita water resources are only 1/4 of the world average, and the distribution is not even. In terms of energy, China’s total energy consumption has been in the forefront of the world for several consecutive years [1]. In 2016, it surpassed the United States and became the world’s largest energy consumer, accounting for 22.6% of the world’s primary energy consumption. In 2018, China’s crude oil imports reached a new high, importing 460 million tons crude oil throughout the year, an increase of 10.1% over the previous year, which shows that China’s energy consumption is extremely large, and it is falling behind in self-sufficiency [2].

At the same time, China’s environment has also paid a huge price—China has become the world’s largest emitter of carbon dioxide, accounting for 28% of the world’s total [2]. According to World Health Organization data, about 4.2 million people died prematurely from particles (PM_2.5_) of air pullutants with a diameter of 2.5 micrometers or less. in 2016, while 58% of people died of ischemic heart disease and stroke due to air pollution; 18% died of chronic obstructive pulmonary disease and acute lower respiratory tract infection; 6% of people died of lung cancer. At the same time, heavy metals in wastewater discharge also bring great hidden dangers to people’s health. The impact of environmental pollution on human health is self-evident. In August 2016, General Secretary Xi Jinping delivered an important speech at the National Health and Wellness Conference, stating: “Put people’s health in a strategic position of priority development” and “effectively solve outstanding environmental problems that affect the health of the people.” In October 2017, the report of the 19th National Congress of the Communist Party of China pointed out the implementation of a healthy China strategy: We must improve the national health policy and provide comprehensive and comprehensive health services for the people. In July 2019, the state formulated and issued the “Healthy China Action (2019–2030)”. Whether it is energy input, economic output or health efficiency, our focus is to adhere to sustainable economic development. Relevant issues have also led to in-depth discussions between scholars and experts.

Data envelopment analysis (DEA) was first proposed by Charnes [3] and others. It is a method for evaluating decision-making units. The DEA method does not need to presuppose the form of production function, and can handle the problems of multiple inputs and outputs, so it has attracted the attention of many scholars at home and abroad. For the relationship between economic development and environmental pollution, Zhang et al. [4] used the environmental Kuznets curve (EKC) hypothesis to study the relationship between rural non-point source pollution and economic development in the Three Gorges reservoir area. This method is limited to the inverse U-shaped relationship between static economy and environment, and does not consider multiple forms of environmental-revenue theory. Peng et al. [5] studied the environmental management efficiency of the Yangtze River urban agglomeration through DEA, a Tobit model and efficiency deconstruction method. This article mainly determines the unfavorable factors and favorable factors in urban construction by further decomposing the efficiency of environmental governance, thus helping urban agglomerations to effectively avoid the adverse effects of environmental governance efficiency on economic development and achieve coordinated development of urban construction and environmental governance. However, the specific impact of these factors has not been thoroughly analyzed. More scholars explore the relationship between energy and air pollution: Choi et al. [6] constructed a DEA to calculate and empirically study the marginal abatement cost of carbon dioxide, and analyze related policies. The environmental investigation and analysis of carbon dioxide alone is not comprehensive enough to objectively reflect the relationship between energy and the environment; Wang et al. [7] used exponential decomposition analysis (IDA) and structural decomposition analysis (SDA) to overlap the largest economic analysis. Study the drivers of energy consumption and energy-related emissions changes and analyze relevant policies. In addition to economic development and environmental, energy consumption and environmental discussion, many scholars have conducted research on air pollution and human health. For example, Pope [8] studies the epidemiology of atmospheric fine particulate pollution and human health; Gurjar et al. [9] use spreadsheets to assess research on atmospheric pollution and human health in large cities; Khaniabadi et al. [10] evaluate the effects of PM10, NO_2_, and O_3_ on human health.

Existing research on energy–economy–environment is mostly carried out separately from the health dimension. For example, energy–economic–environmental research or health–environmental research mostly use multistage research. There is little research that combines productivity and health efficiency, and provides an in-depth analysis of their connections. This study is based on the modified undesirable dynamic network model to study the productivity and health efficiency of China’s 30 provinces (excluding autonomous regions and municipalities, except the Tibet Autonomous Region) for 2014–2017. The contribution of this paper is mainly presented in two aspects. First, in addition to exploring the economic, energy and environmental pollution efficiency, this study also includes the assessment of health efficiency, which can comprehensively explore the four major aspects of economy, energy, environmental pollution and human health. Second, this paper is the first to study the two-stage efficiency of 30 provinces (because of the incomplete data of the Tibet Autonomous Region, so it is excluded) from the national level. It can provide data reference for the country and provinces from both macro and micro aspects. This study is mainly evaluated in two phases. In the first phase, production efficiency, capital labor, energy, water consumption and fixed asset investment are used as input indicators, and GDP is the desired output, and chemical oxygen demand (COD), CO_2_ and chromium emissions are undesired outputs. On this basis, the second phase, the study of health efficiency, and the local financial health expenditure and the number of health technicians are used as input indicators. The health index and the population mortality rate are the agreed output and the undesired output index respectively. Fixed asset investment was selected as the carryover indicator in both phases. Due to the large resource factor endowments among provinces, there are large differences in economic development level, economic industrial structure, geographical environment, climate characteristics, living habits and customs, so each province will have different key issues in the development process. This study compares the production efficiency and health efficiency of 30 provinces (excluding Tibet) in China. First, it observes the total efficiency values and the efficiency values of different stages in each year, and pays attention to their fluctuations, and proves the contribution or drug effect of each stage on the overall efficiency values. Then it analyzes the different problems highlighted by the provinces in terms of economic input and health output from the perspectives of the whole and the local, and circles the key indicators of different provinces by comparing the improvement space between different years, different provinces and different indicators. We observe the changes in their efficiency values and make targeted recommendations for each region. This paper conducts comprehensive research on both production efficiency and health efficiency, aiming at improving energy efficiency while paying attention to health efficiency, and providing a scientific basis for the construction of a healthy China and the sustainable development of the economy.

The rest of the paper is organized as follows: Section 2 is a literature review, Section 3 is a methodology introduction, Section 4 is the result of an empirical analysis, Section 5 summarizes the paper, and proposes relevant recommendations.

## 2. Literature Review

The limited availability of energy was pointed out at the 7th World Conference on Recycling, Recycling and Reintegration. In fact, since as early as the mid-1980s, with the emergence of environmental issues, the issue of energy sustainability has received attention. Well-known universities such as Yale University and Harvard University offer courses in energy economics. At present, the research on energy and economic environment and health mostly focuses on environmental issues or simply studies the sustainable development between energy and economy. Most of them use DEA one-stage research and lack comprehensive research on economy and health. This paper is based on a modified undesirable dynamic network model, adopting a two-stage study of “production phase” and “health phase” to assess the economic input and health output of 30 provinces in China. For different efficiency values of indicators at different stages of each province, we compare their improvement space and propose targeted recommendations for different regions.

According to the previous references, there are few studies on the same level of economic development, energy consumption, environmental pollution and human health, and the analysis of related split combinations mainly has the following aspects.

### 2.1. Research on Economics, Energy and the Environment

Research on economic development and environmental pollution, such as Zhang et al. [4] mentioned above, uses the environmental Kuznets curve (EKC) hypothesis to study the relationship between rural non-point source pollution and economic development in the Three Gorges reservoir area; Peng et al. [5] studied the environmental governance efficiency of the Yangtze River urban agglomeration through DEA, Tobit model and efficiency deconstruction method; Meng et al. [11] studied the relationship between economic growth, technological innovation and air pollution, and found technological innovation is conducive to the change of EKC shape. However, all of the above are based on the static state of the environment from the current state of the environment, corresponding to the economic development of the time, and did not consider the effect of dynamic two-way effects. As energy and pollution are increasingly valued, Chen et al. [12] conducted a detailed study of China’s energy use and related air pollution in recent years, in view of China’s extreme energy consumption and severe environmental pollution. Constructive suggestions were made on the sustainable use of energy and pollution control. In 2009, Fang et al. [13] pointed out that China’s economic development has relied heavily on coal as an energy source, both in the past and now, which has led to serious air pollution. However, due to the particularity of China’s national conditions, the environmental protection model of developed countries does not apply to China. Studies have shown that in the area of environmental protection, we must continue to intensify efforts, but we must also re-examine the traditional “first clean up” approach. Choi et al. [6] used the DEA model to estimate the potential reduction and efficiency of China’s carbon dioxide emissions, and conducted empirical research to discuss the significance of relevant policies. Li and Hu [14] used the slack-based model (SBM) to calculate the energy-efficiency factor (ETFEE) of 30 regions in China from 2005 to 2009, but the results were not satisfactory. The regional energy efficiency in China was overestimated at 0.1 or more. The study also shows that the ratio of the secondary industry to GDP and the government’s subsidies for industrial pollution have a negative impact on the country’s economic development. Zhang et al. [15] proposed a non-radial distance function of the frontier to simulate the carbon dioxide emission performance in power generation, and empirically studied the fossil fuel power generation in Korea. They point out that coal-fired power plants are more efficient than oil-fired power plants. Finally, they suggested that the Korean government should strengthen the technology of fuel-fired power plants to achieve an overall improvement in the efficiency of the power generation industry. Apergis and Ozturk [16] focused on the relationship between national income and emissions (environment), using the generalized method of moments (GMM) method of panel data to test the EKC hypothesis in a multivariate framework, providing empirical support for EKC. Emil Georgiev and Emil Mihaylov [17] also validated the EKC. But they found that the inverted U-shape between income and pollution does not apply to all gases, and a meaningful EKC exists only in carbon monoxide (CO), volatile organic compound (VOC), and oxides of nitrogen (NOX). In 2016, Zhang et al. (2016) [4] first used the environmental Kuznets curve hypothesis to study the relationship between rural pollution and economic development in the Three Gorges area. They also found that EKC does not apply to all pollutant variables, such as agricultural waste production (cereal residues, etc.). Li et al. [18] also based their work on the environmental Kuznets curve theory, plus a panel smooth transition regression model to study the relationship between CO_2_ and SO_2_ representative gas and economic growth. The study also shows that there is no inverted U-shaped relationship between China’s CO_2_ emissions and economic growth, but SO_2_ is consistent. This result, on the basis of verifying international experience, also has certain significance for China’s economic development and environmental emissions. In addition to the application of EKC, Guo et al. [19] used a dynamic DEA model to evaluate energy efficiency in Organisation for Economic Co-operation and Development (OECD) countries and China. Uniquely, based on the assessment of energy stocks, it provides standards for the adjustment of energy stocks. Proposals for countries to increase energy stocks in order to improve energy efficiency are proposed. Qin et al. [20] also used data envelopment analysis to study the energy efficiency of coastal areas in China. The difference is that they propose a global measurement method based on epsilon to estimate static energy efficiency, and use the global Malmquist–Ruinberg productivity index based on the direction distance function to dynamically evaluate energy efficiency, and obtain technological progress. Scale efficiency and the like are the key factors for improving energy efficiency. Wang and Fan et al. [21] first used the gravity model to study the spatial distribution and center of gravity of energy-related CO_2_ emissions, and used the structural effects of the Logarithmic Mean Divisia Index (LMDI) method at the national and provincial levels to find that on multiple spatial scales, the contribution rate of the factor is significantly different between provinces, and the positive driving effect is greater than the negative inhibition effect. Based on these empirical results, policy recommendations for further reduction of CO_2_ emissions are proposed. Li and Xu [22] also conducted research on air pollution in the Yangtze River Delta region of China. Through the development of economy and energy in the Yangtze River Delta region in the past decade, they combined the pollution level to analyze the coupling relationship between air pollution and economic development. Finally, it should be pointed out that the adjustment of energy structure and industrial structure and joint prevention and control of air pollution are key steps in pollution control. Li et al. [23] used 2013–2016 energy consumption data from 31 Chinese cities to evaluation the dynamic efficiency of the urban environments. Labor, fixed assets, and energy consumption were taken as the inputs, gross domestic product (GDP) was taken as the output, and particulate matter (PM_2.5_) and sulfur dioxide (SO_2_) were taken as the carry-over variable indicators. Using a meta-frontier dynamic DEA model, the 31 cities were classified into high-income and upper-middle-income cities, the overall 2013–2016 energy consumption and air pollutant efficiency scores were analyzed, and improvements and changes were recommended to increase the efficiencies. The needed improvements with the high-income cities performing better overall than the upper-middle-income cities. While there have been some significant improvements in SO_2_ emissions, PM_2.5_ improvements have been far slower. Li et al. [24] measured energy efficiency, carbon dioxide emissions efficiency, and particulate matter (PM_2.5_) concentration efficiency to compare the energy efficiency differences between OECD member countries and non-OECD member countries from 2010 to 2014 using a metafrontier dynamic Data Envelopment Analysis model. They calculated technology gap ratio and input and output efficiency values to measure the energy efficiencies of each economy, finding that OECD countries have a technology gap ratio of 1 or very close to 1; and except for the United Arab Emirates and Singapore, both of which exhibit annual improvements, the non-OECD countries have a significant need for efficiency improvements.

### 2.2. Research on Water Resources, Environment and Human Health

It is difficult to imagine a modern society without the benefits of chemicals and the chemical industry. Harrison. [25] considered that pharmaceuticals, petrochemicals, agrochemical, industrial and consumer chemicals all contribute to our modern lifestyles. However, with the rise of chemical manufacture and use has come increasing public awareness and concern regarding the presence of chemicals in the environment. Inadequate quantity and quality of surface water resources in China have led to the long-term use of waste-water irrigation to fulfill the water requirements for agricultural production. In some regions this has caused serious agricultural land and food pollution, especially for heavy metals. Therefore, Lu et al. [26] argued that issues threatening food safety such as combined pesticide residues and heavy metal pollution should be addressed to reduce risks to human health. Lee et al. [27] conducted a health survey of three villages near a pulp and paper mill along the Kampa River in Riau province, Indonesia. Through the analysis of river water and the study of the skin conditions of children in different villages in the upper and lower reaches, they found that, except for the fecal *E. coli* population, wastewater from the factory is less likely to cause skin conditions within the physical and chemically acceptable range. Most of the diarrhea that occurs is caused by *E. coli* because all untreated sewage is deposited in the river. Wu and Sun [28] also evaluated groundwater pollution and health effects in central and western China. According to research, more than 60% of the water in the area is not suitable for drinking. Most of the water is generally suitable for irrigation, and oral intake is the main exposure route for health risks. Therefore, it is necessary to take urgent and effective measures to prevent groundwater pollution and reduce the health risks in the area. Remoundou and Koundouri [29] provides a review of the literature on valuation studies eliciting monetary values associated with reduced environmental risk, in particular focusing on reduced indoor and outdoor air pollution, enhanced water quality and climate change mitigation. “Discharge of treated wastewater and sewage sludge land-filling” are the most common practices, which pose threats to the local environment. Zhang et al. [30] constructed a sewage treatment ecosystem, including a sewage treatment system, treated water disposal system and dewatered sludge disposal system to improve the economic and environmental impacts of waste treatment and disposal. Sun et al. [31] conducted a study on water treatment plants. They used the cluster variation (CV) method framework to evaluate the value of the water treatment plant and then established a binary sample selection model to reduce selectivity errors. Through research, they found that compared with the pollution discharge of water treatment plants, the public is more willing to improve efficiency, increase governance, reduce pollution to pay for water treatment plants, and pay for health problems. Pouriyeh et al. [32] used the DEA method to measure the efficiency of urban development Yazd City, Central Iran. The present study is an attempt to evaluate the performance and efficiency of development of Yazd City using the DEA over the years 1983–2013. In this regard, the ecological factors, affecting the growth of the city of Yazd in the study period, were identified initially. The factors include elevation, slope, aspect, geology, morphology, soil, water quantity, climatic features, and land cover. Next, using variable returns to scale (Banker/Charnes/Cooper, BCC) based on the output-oriented approach, the efficiency of development of Yazd City was calculated by the general algebraic modeling system (GAMS) software to recognize efficient and inefficient units. Then, Anderson-Peterson (AP) ranking method was used to rank the most efficient units in the development of Yazd City over the study years. Saha et al. [33] conducted an investigation of the surface and groundwater in the Dhaka Export Processing Zone and assessed the health risks of the people in the area through a definitive and probabilistic approach. Studies have shown that industrial emissions of heavy metals and other pollutants make water quality far below the safety limit, and the risk of exposure to cancer in adults is higher. Yan et al. [34] evaluated the spatiotemporal pattern of PM_2.5_ concentrations in: China—A case study from the Beijing-Tianjin-Hebei region. The study considers the spatiotemporal characteristics, and believes that the spatial pattern of PM_2.5_ is related to the season. The PM_2.5_ concentration in Beijing-Tianjin-Hebei has obvious spatial spillover effects. Rybaczewska-Blazejowska, Magdalena and Masternak-Janus, Aneta assessed Polish regions in DEA: Joint application of life-cycle assessment and data envelopment analysis. The combined application of life-cycle assessment (LCA) and DEA—the input-oriented BCC model—has been chosen as a tool for the comprehensive eco-efficiency assessment, due to its high capability to measure regional eco-efficiency. The ultimate goal of this approach is to support the strategic decision-making process.

### 2.3. Research on the Economy, Environment and Human Health

The industrialization of early economic development was at the expense of environmental pollution, and the average life expectancy of the population was low. As mentioned above, Pope, CA [8] briefly summarizes the epidemiology of industry’s impact on air pollution, which has an impact on health. He pointed out that in the short-term acute high exposure to fine-grained pollution, the elderly and babies are the most vulnerable. Gurjar, B.R. et al. [9] also assess health risks from the perspective of airborne morbidity and mortality. It has also been found that cardiovascular mortality is highest in environments with high concentrations of suspended particulate matter (TSP), and other chronic disease rates follow this trend. It can be seen that improving air quality and preventing high levels of air pollution and causing excessive mortality and morbidity are important measures for environmental governance and human health. Rezaee et al. [35] used DEA and bargaining game model for evaluation of health centers. In other words, two categories of measures are used to measure unified efficiency for each health center in the competitive environment. Two models according to constant return to scale (CRS) and variable return to scale (VRS) assumptions are developed. Health centers as an important part of the healthcare systems are considered for evaluation. Although there is a growing interest in climate change and health research, specific in-depth studies are rare, so Herlihy et al. [36] examine the literature on the relationship between global climate change and human health to define the scope, to provide data for future understanding of the impact of climate change on health. Wu and Sun [28] studied the effects of shallow groundwater pollution on alluvial plains on human health under the influence of agricultural and industrial activities in the central and western regions of China. Saha et al. [33] conducted a probabilistic assessment of industrial metal pollution and human health risks in water, and the results showed that the deterministic and probabilistic estimates of cancer risk through exposure to groundwater were well below the safety limits. Liang et al. [37] studied human health from the perspective of consumption. As globalization progresses, international trade is increasingly popular around the world, but it separates the consumption of goods and services from its production and production-based environmental emissions. Based on this, scholars quantified the global economic output caused by consumption in 189 countries around the world and the health effects of human health related to PM_2.5_. The study found that when developing countries outsourced economic production to Asian countries, they also outsourced PM_2.5_ to health-related factors. To this end, joint efforts between developed and Asian countries are needed to reduce the global human health impact associated with PM_2.5_. As globalization progresses, developed countries outsource production and make developing countries’ economic development more rapid, but at the same time they are at the expense of their environment and the impact of people’s health. Air pollution in developing countries is generally more serious. In 2017, Khaniabadi et al. [10] and others conducted research on the Iranian region. They pointed out that air pollution has become a risk factor for people’s health. The excess of cardiovascular mortality is estimated by the relative risk (RR) and baseline incidence (BI) defined by the World Health Organization (WHO). They suggested that adopting policies and actions to reduce the various sources of these pollutants in transportation and energy manufacturing facilities is an important measure to reduce the status quo and reduce hazards. Chen et al. [38] based on the PM_2.5_ of the China, Children, Family and Health (CCHH) program to study the effects of asthma and allergic diseases or symptoms in preschool children in six cities, long-term exposure to PM_2.5_ may increase the risk of asthma and allergic diseases or symptoms in Chinese preschool children. Children living in suburban or rural areas are at higher risk of exposure to PM_2.5_ than children living in urban areas. Zhu et al. [39] studied the emissions of polycyclic aromatic hydrocarbons in China. They found that compared with developed countries, China’s polycyclic aromatic hydrocarbon emissions are at a relatively high level. They provide a scientific basis for the government to formulate relevant emission reduction measures through a comprehensive review of the emissions of polycyclic aromatic hydrocarbons and health risks. Excluding the environmental pollution brought about by industrialization and the environmental and health impacts brought by economic globalization to developing countries, economic development and the improvement of people’s living standards have also had certain effects on human health, such as interior decoration. In 2013, Yun et al. [40] found that excessive indoor decoration caused serious indoor formaldehyde and total volatile organic compounds (TVOC) pollution, and proposed some measures to prevent indoor air pollution. Yang and Li. [41] Based on existing academic research in China and abroad, as well as with the help of the Data Envelopment Analysis (DEA) model, adopted 39 industrial sectors of China as decision-making units (DMU) and their input-output data from 2003 to 2014 to calculate and evaluate the tetrafluoroethylene (TFE) of waste gas control in different industries of China. By designing an original MATLAB algorithm for optimization solutions of multi-variable constrained nonlinear functions, they found that there were 24 industries (61.5% of total industries) whose waste gas control efficiency at period end was lower than that at period beginning. Xu et al. [42] used a hybrid of panel data analysis and an augmented DEA, modeled human resources, material, finance to determine their technical and scale efficiency to comprehensively evaluate the transverse and longitudinal allocation efficiency of community health resources in Jiangsu Province. They observed that the Deepen Medical Reform in China has led to an increase in concern to ensure efficient allocation of community health resources by health policy makers in the province. This has led to greater efficiency in health resource allocation in Jiangsu in general but serious regional or municipal disparities still exist.

Current scholars’ research on energy use focuses on the environmental pollution caused by industrial use or the impact on human health. In energy and environmental pollution, scholars use DEA dynamic models, SBM and other models to assess carbon dioxide emissions and ecological environment. Furthermore, scholars have validated the environmental Kuznets curve hypothesis. They found that the consumption of energy such as coal is particularly serious for air pollution, and the emissions of ozone, SO_2_, etc., which are important for the formation of severe weather such as acid rain, are not applicable to the environmental Kuznets curve assumption. This curve only makes sense in CO, VOC and NOX. In other gases, the gas and income are not as inverted as the “U” type. Through these studies, scholars have also put forward some suggestions on the efficient use of energy, such as the adjustment of energy structure and industrial structure. In addition, because most of the energy is not renewable, increasing energy stock is also an important measure. Human life is inseparable from water, therefore, in addition to energy and economic research, the relationship between water resources and human health is also highly valued by scholars. Scholars and experts have conducted many surveys of groundwater or water treatment plants in some industrial areas, and evaluated the health risks of nearby people through CV and binary samples. They found that some of the heavy metals and other germs that are often found in sewage expose people to high risk of carcinogenesis. Health problems are becoming more and more prominent. People are more willing to pay for their own health than the economic growth brought about by industrial development. The process of globalization has made industrial emissions in developing countries even worse, and some suspended matter in the atmosphere has made chronic diseases aggravate the possibility of concurrency. Although scholars have many discoveries and achievements in energy and economy, there is still a lack of research that combines production efficiency with health efficiency.

How to use energy efficiently, effectively prevent pollution, and maintain economic sustainability are still issues that need to be resolved. The non-renewable energy, maintaining economic sustainability and ensuring human health are things that must be addressed to advance human development. At present, the economic development processes and the industrial structure of various provinces in China are different, and the advanced nature of technology is not uniform. Local governments pay different attention to health issues. How to formulate guidelines for different situations in different provinces, making production efficiency and health efficiency parallel is the purpose of this study, and also the significance of our research.

This paper is based on existing research on the utilization of energy sources and health inputs. Data from 30 provinces (except Tibet) were processed through the DEA two-stage dynamic evaluation model. In the first stage, that is, the production efficiency stage, we introduce energy consumption standard coal, employment population, etc., and output variables such as GDP, CO_2_, COD, etc., on the basis of which the second stage—health efficiency research, but joined Inputs such as chromium emissions, number of health technicians, and local fiscal health expenditures output new variables such as health index and population mortality. The output variables are further divided into desirable and non-conforming indicators. According to the research results, we can know that most provinces have productivity values higher than health efficiency values, and health efficiency has a drag on overall efficiency. It shows that most provinces in China pay special attention to economic output, while for relative health problems, there is less investment. For CO_2_, COD and other emissions, they are found to have a great relationship with the utilization of water resources. In order to prevent this pollution, updating energy and water technology should be the focus. In addition, we also concluded that the provinces with higher GDP generally have higher efficiency. But productivity and health efficiency are sometimes related to the geography. In some areas with small populations, production is not developed, but health and efficiency are high; but some resource provinces have higher production efficiency. For provinces with high production efficiency, health investment should be increased, and the training of medical personnel and the guarantee of medical facilities should be emphasized. We must also pay attention to the reduction of emissions and the improvement of water-use efficiency. For some provinces with relatively backward development but good health efficiency, we attach importance to the adjustment of industrial structure in the region, update energy-related technologies and rationally cultivate talents to improve the overall quality of each talent with the industry counterpart. This paper provides a scientific basis for government decision-making, improving resource utilization, reducing health problems, and promoting sustainable economic development by linking the province’s GDP, geographical environment and other factors and their different performances at various stages.

## 3. Research Methods

Farrell [43] first proposed the concept of measuring efficiency by production frontier, and divided production efficiency into technical efficiency and allocative efficiency. The follow-up scholars continue to discuss according to the Farrell theory. The main theoretical basis is the 1978 CCR (Charnes, Cooper and Rhodes) model and the 1984 BCC (Banker, Charnes and Cooper) model. Since the CCR model and the BCC models measure the radial efficiency, that is, the input or output items can be increased or decreased in equal proportions, however, this assumption is not applicable in all cases. Therefore, Tone [44] proposed the slack-based measure (SBM) in 2001, using the difference variable as the basis for measurement, taking into account the difference between the input and output items (slack), and non-ray (non-Radial) is estimated in a scalar manner to represent SBM efficiency.

After Tone proposed the SBM, DEA theory continued to develop. In 2007, Färe et al. [45] proposed network data envelopment analysis (NDEA), which believed that the production process was composed of many the sub-production technology is formed, and this production technology is regarded as the sub-decision unit (Sub-DMU), and the optimal solution is obtained by the traditional CCR and BCC modes. Compared with the traditional DEA mode, this production technology is recognized as a “black box”. The network DEA mode no longer regards it as a black box that cannot be included in the evaluation. Instead, it applies these production technologies to discuss the impact of the input configuration on the production process.

Following Färe et al., Tone and Tsutsui [46] further proposed a weighted slack-based measures network data envelopment analysis model in 2009. The connection between the various departments of the decision-making unit is used as the basis for the analysis of the network DEA model, and each department is regarded as Sub-DMU to find the optimal solution by using the SBM model. The Network DEA model improves the traditional DEA’s failure to analyze the performance of each department, but fails to consider the time factor. Subsequent developments were by Malmquist [47], Fare et al. [48], and the Malmquist Index, which is a model for measuring changes in intertemporal efficiency, but these models do not consider intertemporal continuation. The effect of the activity is not suitable for measuring long-term efficiency. Fare and Grosskopf [49] were the first to use the carry-over to put connected variables into a dynamic model, and Tone and Tsutsui [50] extended the model to a dynamic analysis of the slacks-based measure. In 2014, Tone and Tsutsui [51] proposed weighted slack-based measures (dynamic network DEA) data envelopment analysis mode, using the connectivity between the various departments of the decision-making unit as the basis for the analysis of the network DEA model. Each department is considered a Sub-DMU, and carry-over activities are used as a link. Carry-over activities can be divided into four categories, (1) desirable (good), (2) random (bad), (3) discretionary (changeable), (4) non-discretionary (Unchangeable), as the basis for the analysis of the dynamic DEA model; and the DEA model is divided into three forms, namely, input-oriented, output-oriented and non-directed, and then use the SBM model to find the optimal solution.

In the first stage, the production stage, the labor, energy and water consumption are used as input indicators, GDP is the desired output. COD, CO_2_ and chromium emissions are undesired output indicators; in the second stage, the health stage, the local financial health expenditure and the number of health technicians are used as input indicators. The health index and the population mortality rate are the agreed output and the undesired output index respectively. Fixed asset investment was selected as the carryover indicator in both phases.

Since this study considers undesirable output in the dynamic network SBM model, we can modify Tone and Tsutsui’s dynamic network model to be the modified undesirable dynamic network model. The modified undesirable dynamic network model, was designed as follows:

Suppose there are *n DMU*s (*j* = 1, …, *n*), with each having *k* divisions (*k* = 1, …, *K*), and *T* time periods (*t* = 1, …, *T*). Each of the *DMU*s has an input and output at time period *t* and a carryover (link) to the next *t* + 1 time period.

Set $m_{k}$ and $r_{k}$ to represent the input and output in each division *K*, with (k,h)i representing divisions *K* to *h* and *L_hk_* being the *k* and *h* division set. The input and output, links, and carryover definitions are given in the following.

Inputs and outputs: $X_{ijk}^{t}\in R_{+}^{}$$\left(i=1,\ldots ,m_{k}^{};\;j=1,\ldots ,n;K=1\ldots ,K;t=1,\ldots ,T\right)$ refers to input *i* at time period *t* or *DMU_j_* division *k*. $y_{rjk}^{t}\in R_{+}^{}$$\left(r=1,\ldots ,r_{k}^{};\;j=1,\ldots ,n;K=1\ldots ,K;t=1,\ldots ,T\right)$ refers to output *r* in time period t for *DMU_j_* division *k*; if part of the output is not ideal, it is considered an input for the division.

Links: $Z_{j(kh)t}^{t}\in R_{+}^{}$$\left(j=1;\ldots ;n;l=1;\ldots ;L_{hk}^{};t=1;\ldots ;T\right)$ refers to the period *t* links from *DMU_j_* division k to division h, with $L_{hk}^{}$ being the number of k to h links.

Carryovers: $Z_{jkl}^{\left(t,t+1\right)}\in R_{+}^{}$
$\left(j=1,\ldots ,n;l=1,\ldots ,L_{k}^{};k=1,\ldots k,t=1,\ldots ,T-1\right)$ refers to the carryover of t to the t+1 period from $DMU_{j}$ division k to division h, with $L_{k}^{}$ being the number of carryover items in division k. The following is the non-oriented model:
(a)Objective functionOverall efficiency:
(1)$$\theta_{0}^{*}=\min\frac{\sum_{t=1}^{T}W^{t}\left[\sum_{k=1}^{K}W^{k}\left[1-\frac{1}{m_{k}+linkin_{k}+ninput_{k}}(\sum_{i=1}^{m_{k}}\frac{S_{iok}^{t-}}{x_{iok}^{t}}+\sum_{{\left(kh\right)}_{l}=1}^{linkin_{k}}\frac{s_{o{\left(kh\right)}_{l}in}^{t}}{z_{o{\left(kh\right)}_{l}in}^{t}}+\sum_{k_{l}}^{ninput_{k}}\frac{s_{ok_{l}input}^{\left(t,t+1\right)}}{z_{ok_{l}input}^{\left(t,t+1\right)}})\right]\right]}{\sum_{t=1}^{T}W^{t}\left[\sum_{k=1}^{K}W^{k}\left[1+\frac{1}{r_{1k}+r_{2k}}(\sum_{r=1}^{r_{1k}}\frac{s_{rokgood}^{t+}}{y_{rokgood}^{t}}+\;\sum_{r=1}^{r_{2k}}\frac{s_{rokbad}^{t-}}{y_{rokbad}^{t}})\right]\right]}$$Subject to:
(2)$$x_{ok}^{t}=X_{k}^{t}\lambda_{k}^{t}+s_{ko}^{t-}\left(\forall k,\forall t\right)$$
(3)$$y_{okgood}^{t}=Y_{kgood}^{t}\lambda_{k}^{t}-s_{kogood}^{t+}\left(\forall k,\forall t\right)$$
(4)$$y_{okbad}^{t}=Y_{kbad}^{t}\lambda_{k}^{t}+s_{kobad}^{t-}\left(\forall k,\forall t\right)$$
(5)$$e\lambda_{k}^{t}=1\left(\forall k,\forall t\right)$$
(6)$$\lambda_{k}^{t}\geq 0,s_{ko}^{t-}\geq 0,s_{kogood}^{t+}\geq 0,s_{kobad}^{t-}\geq 0,\left(\forall k,\forall t\right)$$
(7)$$Z_{o(kh)in}^{t}=Z_{(kh)in}^{t}\lambda_{k}^{t}+S_{o(kh)in}^{t}\left((kh)in=1,\ldots ,linkin_{k}\right)$$
(8)$$\sum_{j=1}^{n}z_{jk_{1}\alpha }^{\left(t,(t+1)\right)}\lambda_{jk}^{t}=\sum_{j=1}^{n}z_{jk_{1}\alpha }^{\left(t,(t+1)\right)}\lambda_{jk}^{t+1}\left(\forall k;\forall k_{l};t=1,\ldots ,T-1\right)$$
(9)$$Z_{ok_{l}input}^{\left(t,(t+1)\right)}={\sum_{j=1}^{n}z}_{jk_{l}input}^{\left(t,(t+1)\right)}\lambda_{jk}^{t}+s_{ok_{l}input}^{\left(t,(t+1)\right)}k_{l}=1,\ldots ,ngood_{k};\forall k;\forall t)$$
(10)$$s_{ok_{l}good}^{\left(t,(t+1)\right)}\geq 0,\left(\forall k_{l};\forall t\right)$$(b)Period and division efficienciesPeriod and division efficiencies are as follows:
(b1)Period efficiency:
(11)$$\partial_{0}^{*}=\min\frac{\sum_{k=1}^{K}W^{k}\left[1-\frac{1}{m_{k}+linkin_{k}+ngood_{k}}(\sum_{i=1}^{m_{k}}\frac{S_{iok}^{t-}}{x_{iok}^{t}}+\sum_{{\left(kh\right)}_{l}=1}^{linkin_{k}}\frac{s_{o{\left(kh\right)}_{l}in}^{t}}{z_{o{\left(kh\right)}_{l}in}^{t}}+\sum_{k_{l}}^{ngood_{k}}\frac{s_{ok_{l}good}^{\left(t,t+1\right)}}{z_{ok_{l}good}^{\left(t,t+1\right)}})\right]}{\sum_{k=1}^{K}W^{k}\left[1+\frac{1}{r_{1k}+r_{2k}}(\sum_{r=1}^{r_{1k}}\frac{s_{rokgood}^{t+}}{y_{rokgood}^{t}}+\;\sum_{r=1}^{r_{2k}}\frac{s_{rokbad}^{t-}}{y_{rokbad}^{t}})\right]}$$(b2)Division efficiency:
(12)$$\varphi_{0}^{*}=\min\frac{\sum_{t=1}^{T}W^{t}\left[1-\frac{1}{m_{k}+linkin_{k}+ninput_{k}}(\sum_{i=1}^{m_{k}}\frac{S_{iok}^{t-}}{x_{iok}^{t}}+\sum_{{\left(kh\right)}_{l}=1}^{linkin_{k}}\frac{s_{o{\left(kh\right)}_{l}in}^{t}}{z_{o{\left(kh\right)}_{l}in}^{t}}+\sum_{k_{l}}^{ninput_{k}}\frac{s_{ok_{l}input}^{\left(t,t+1\right)}}{z_{ok_{l}input}^{\left(t,t+1\right)}})\right]}{\sum_{t=1}^{T}W^{t}\;\left[1+\frac{1}{r_{1k}+r_{2k}}(\sum_{r=1}^{r_{1k}}\frac{s_{rokgood}^{t+}}{y_{rokgood}^{t}}+\;\sum_{r=1}^{r_{2k}}\frac{s_{rokbad}^{t-}}{y_{rokbad}^{t}})\right]}$$(b3)Division period efficiency:
(13)$$\rho_{0}^{*}=\min\frac{1-\frac{1}{m_{k}+linkin_{k}+ninput_{k}}(\sum_{i=1}^{m_{k}}\frac{S_{iok}^{t-}}{x_{iok}^{t}}+\sum_{{\left(kh\right)}_{l}=1}^{linkin_{k}}\frac{s_{o{\left(kh\right)}_{l}in}^{t}}{z_{o{\left(kh\right)}_{l}in}^{t}}\sum_{k_{l}}^{ninput_{k}}\frac{s_{ok_{linput}input}^{\left(t,t+1\right)}}{z_{ok_{l}input}^{\left(t,t+1\right)}})}{1+\frac{1}{r_{1k}+r_{2k}}(\sum_{r=1}^{r_{1k}}\frac{s_{rokgood}^{t+}}{y_{rokgood}^{t}}+\;\sum_{r=1}^{r_{2k}}\frac{s_{rokbad}^{t-}}{y_{rokbad}^{t}}+)}$$

From the above, the overall efficiency, period efficiency, division efficiency and division period efficiency were obtained for 31 Chinese cities from 2014–2017.

Energy consumption, water consumption, GDP, COD, CO_2_ and chromium emissions, health expenditure, financial health expenditure, the number of health technicians, health index and population mortality rate.

We use total-factor energy efficiency index to overcome any possible bias in the traditional energy efficiency indicator. There are 11 key features of this present study: energy consume, water consumption, GDP, COD, CO_2_ and chromium emissions, health expenditure, the number of health technicians, health index and population mortality rate. In our study, “I” represents area and “t” represents time.
(14)$$\mathrm{Energy}\;\mathrm{consumption}\;\mathrm{efficiency}=\frac{\mathrm{Target}\;\mathrm{Energy}\;\mathrm{input}(i,t)}{\mathrm{Actual}\;\mathrm{Energy}\;\mathrm{input}(i,t)}$$
(15)$$\mathrm{Water}\;\mathrm{consumption}\;\mathrm{efficiency}=\frac{\mathrm{Target}\;\mathrm{water}\;\mathrm{input}(i,t)}{\mathrm{Actual}\;\mathrm{water}\;\mathrm{input}(i,t)}$$
(16)$$\mathrm{GDP}\;\mathrm{efficiency}=\frac{\mathrm{Actual}\;\mathrm{GDP}\;\mathrm{desirable}\;\mathrm{output}(i,t)}{\mathrm{Target}\;\mathrm{GDP}\;\mathrm{desirable}\;\mathrm{output}(i,t)}$$
(17)$$\mathrm{COD}\;\mathrm{efficiency}=\frac{\mathrm{Target}\;\mathrm{COD}\;\mathrm{Undesirable}\;\mathrm{output}(i,t)}{\mathrm{Acutal}\;\mathrm{COD}\;\mathrm{Undesirable}\;\mathrm{output}(i,t)}$$
(18)$${\mathrm{CO}}_{2}\;\mathrm{efficiency}=\frac{\mathrm{Target}\;\mathrm{CO}2\;\mathrm{Undesirable}\;\mathrm{output}(i,t)}{\mathrm{Actual}\;\mathrm{CO}2\;\mathrm{Undesirable}\;\mathrm{output}(i,t)}$$
(19)$$\mathrm{Chronmium}\;\mathrm{emissions}\;\mathrm{efficicency}=\frac{\;\mathrm{Target}\;\mathrm{chromium}\;\mathrm{emissions}\;\mathrm{undesirable}\;\mathrm{output}(i,t)}{\;\mathrm{Actual}\;\mathrm{chromium}\;\mathrm{emissions}\;\mathrm{undesirable}\;\mathrm{output}(i,t)}$$
(20)$$\mathrm{Health}\;\mathrm{expenditure}\;\mathrm{efficiency}=\frac{\mathrm{Target}\;\mathrm{health}\;\mathrm{expenditure}\;\mathrm{input}(i,t)}{\mathrm{Actual}\;\mathrm{health}\;\mathrm{expenditure}\;\mathrm{input}(i,t)}$$
(21)$$\mathrm{Number}\;\mathrm{of}\;\mathrm{health}\;\mathrm{technical}\;\mathrm{personnel}\;\mathrm{efficiency}=\frac{\mathrm{Target}\;\mathrm{number}\;\mathrm{of}\;\mathrm{health}\;\mathrm{technical}\;\mathrm{personnel}\;\mathrm{input}(i,t)}{\mathrm{Actual}\;\mathrm{number}\;\mathrm{of}\;\mathrm{health}\;\mathrm{technical}\;\mathrm{personnel}\;\mathrm{input}(i,t)}$$
(22)$$\mathrm{Health}\;\mathrm{index}\;\mathrm{efficiency}=\frac{\mathrm{Target}\;\mathrm{health}\;\mathrm{index}\;\mathrm{efficiency}\;\mathrm{input}(i,t)}{\mathrm{Acutal}\;\mathrm{health}\;\mathrm{index}\;\mathrm{efficiency}\;\mathrm{input}(i,t)}$$
(23)$$\mathrm{Population}\;\mathrm{mortality}\;\mathrm{rate}\;\mathrm{efficiency}=\frac{\mathrm{Target}\;\mathrm{population}\;\mathrm{mortality}\;\mathrm{rate}\;\mathrm{undesirable}\;\mathrm{output}(i,t)}{\mathrm{Acutal}\;\mathrm{population}\;\mathrm{mortality}\;\mathrm{rate}\;\mathrm{undesirable}\;\mathrm{output}(i,t)}$$

If the target energy consumption, water consumption, the number of health technicians and health expenditure input equals the actual input, then the energy consumption, water consumption, the number of health technicians and health expenditure efficiencies equal 1, indicating overall efficiency. If the target energy consumption water consumption, the number of health technicians and health expenditure input is less than the actual input, then the energy consumption water consumption, the number of health technicians and health expenditure efficiencies are less than 1, indicating overall inefficiency.

If the target GDP and health index desirable output is equal to the actual GDP and health index desirable output, then the GDP and health index efficiency equals 1, indicating overall efficiency. If the actual GDP and health index desirable output is less than the target GDP and health index desirable output, then the GDP and health index efficiency is less than 1, indicating overall inefficiency.

If target COD, CO_2_, chromium emissions and population mortality rate undesirable output is equal to the actual COD, CO_2_, chromium emissions and population mortality rate undesirable output level, then COD, CO_2_, chromium emissions and population mortality rate efficiency equals 1, and is efficient. If target COD, CO_2_, chromium emissions and population mortality rate undesirable output is less than the actual COD, CO_2_, chromium emissions and population mortality rate undesirable output level, then COD, CO_2_, chromium emissions and population mortality rate efficiency is less than 1, and is inefficient.

## 4. Empirical Analysis

The following is an empirical analysis of the two-stage dynamic DEA model in various provinces of China.

### 4.1. Data and Variables

This article takes 30 provinces (including autonomous regions and municipalities directly under the Central Government) as the research object, excluding Hong Kong, Macao and Taiwan. Because most of Tibet’s data is missing, it is not included. The production efficiency and health efficiency of the study subjects were evaluated based on a two-stage dynamic DEA model.

This paper considers the different situations in each region. In the first phase, the production phase, capital labor, energy, water consumption and fixed asset investment are used as input indicators. GDP is the desired output, and COD, CO_2_ and chromium emissions are undesired output indicators; in the second phase, the health phase, the local financial health expenditure and the number of health technicians are used as input indicators, and the health index and the population mortality rate are the agreed output and the undesired output index, respectively. Fixed asset investment is selected as the carryover indicator in both phases. Figure 1 shows the framework for the network dynamic model for production and human health efficiency measurements and variables.

The data of energy consumption, CO_2_ (calculated by raw coal, coke, crude oil, gasoline, kerosene, diesel, fuel oil, natural gas, electricity) are extracted from the China energy statistics yearbook (2014–2017); the data of employment population, GDP, fixed assets investment are from the China Statistical Yearbook (2014–2017); the data of water consumption, COD and chromium emissions are from the China water resources bulletin (2014–2017) while the data of financial health expenditure, the number of health technicians, health index (calculated by hospital admission rates) and population mortality rate are from the China health statistics yearbook (2014–2017).

The details of the indicators, as shown in Table 1, are described as follows: ① Labor input. It is expressed by the total number of employed population in each province. ② Energy input. Study on the total energy consumption expressed by standard coal. ③ Water consumption. Refers to the amount of water used by all types of water users, including water loss. ④ Capital investment. Considering the “permanent inventory method” of depreciation of capital stock, depreciation is obtained by accounting for capital stock and capital services, and the study sets the depreciation rate to 0.96. The formula for the perpetual inventory method is expressed as:(24)$$K_{it}=K_{i,t-1}(1-S_{it})+I_{it}$$
*K_it_*, *K_i,t_*_−1_ are the investment stock of this year and the stock of investment of the previous year, and Sit represents the economic depreciation rate of the year. ⑤ GDP. In order to eliminate the impact of price changes, the province’s nominal GDP was converted to real GDP based on the year 2000. ⑥ Carbon dioxide emissions. Because the official data on carbon dioxide emissions is not published, it is calculated by reference to the standard formula published by the Intergovernmental Panel on Climate Change (IPCC).

(25)$$EC=\sum_{i=1}^{n}EC_{i}=\sum_{i=1}^{n}E_{i}CF_{i}\times CC_{i}COF_{i}\frac{44}{12}=\sum_{i=1}^{n}\frac{44}{12}\alpha_{i}E_{i}$$*EC* represents the total amount of carbon dioxide emissions of various types of energy; *i* represents the type of energy consumption. The main sources of energy measured here include 9 kinds, namely raw coal, coke, crude oil, gasoline, kerosene, diesel, fuel oil, natural gas, electricity; *E_i_* represents the consumption of type *i* energy; *CF_i_* is the lowest calorific value; *CC_i_* is the carbon content; *COF_i_* oxidation factor; *α_i_* represents carbon emission coefficient of type *i* energy; 44 and 12 are the molecular or atomic masses of CO_2_ and C, respectively. The internationally accepted IPCC data is shown in Table 2, and the unit is tc/tce. ⑦ Chemical oxygen demand (COD). Refering to the sum of COD emissions from industrial waste water and living waste water, Chemical oxygen demand refers to the amount of oxygen required to oxidize organic contaminants in water with chemical oxidants. The higher the COD value, the heavier the organic pollutants in the water. ⑧ Chromium emissions. During the production process, a large number of heavy metals will flow into the water resources, causing residents to be poisoned or in a sub-health state. This paper selects chromium emissions as a representative of heavy metals emissions. ⑨ The number of health technicians. Health technicians include health practitioners such as practicing doctors, practicing assistant physicians, registered nurses, pharmacists (scientists), inspection technicians (scholars), imaging technicians, health supervisors, and trainees (medicine, nursing, and technical), which represents the second phase of labor input. ⑩ Medical and health expenditure. That is, the medical and health expenditure items in the general budgetary expenditure of local finance. This referred to government spending on health care, specifically, it includes medical and health management affairs expenditures, medical service expenditures, medical insurance expenditures, disease prevention and control expenditures, health supervision expenditures, maternal and child health expenditures, and rural health expenditures. ⑪ Health index. This article uses the percentage of non-hospitals as the health index, and the number of hospitalizations comes from the provincial health and family planning statistics yearbook. ⑫ Population mortality rate. Refers to the ratio of the number of deaths in a certain area to the average number of people in the same period (or interim numbers), expressed in thousands of points. The mortality rate in this data refers to the annual mortality rate, which is calculated as: mortality = annual death population/annual average population * 1000‰

### 4.2. Statistical Analysis of Input and Output

This study selects the input-output data of 30 provinces and cities in China from 2014 to 2017. Throughout the annual statistical data, it can be found that the numbers of employed population, fixed asset investment, GDP, health technicians and local fiscal expenditure indicators have been presenting a rising trend; little changes in energy consumption, total water use, carbon dioxide emissions and population mortality; COD emissions, chromium emissions and health index indicators have shown a slight downward trend. As shown in Table 3, as a result of the input of the production stage, the average number of employed population and energy consumption are increasing year by year. The maximum number of employed population has increased obviously, from 10.1202 million in 2014 to 14.2613 million in 2017, an increase of about 1.3 million per year, while the minimum has increased slowly, with a minimum of 148,700 from 2014 to 2017. The standard deviation of employment population increases slowly year by about 30 but the value of every year is large, which is closely related to the population, economic development level and industrial structure of each province. The increase of the maximum and minimum energy consumption is not obvious, but the maximum value fluctuates. The maximum consumption for the first three years has grown more slowly, with a rise of 14.344 million tons from 2014 to 2015 and a rise of 7.383 million tons from 2015 to 2016, but the maximum consumption for 2017 is lower than the maximum consumption in 2016.

The average amount of water used remained roughly unchanged from 2014 to 2017, and the maximum value fluctuated. The maximum value in 2015 and 2016 was 57.72 billion cubic meters and 57.74 billion cubic meters, which was lower than the 2014 maximum of 59.129 billion cubic meters. In 2017, it rose back to the maximum level in 2014. From the perspective of investment in fixed assets, the average value from 2014 to 2017 will increase by about 1.16 billion yuan per year, and the minimum value will increase by about 300 billion yuan and 230 billion yuan each year.

From the output indicators of the production stage, the maximum GDP has grown steadily since 2014. The maximum value increased by 500.27 billion yuan from 2014 to 2015, and the annual maximum increased by 800 billion yuan in the next two years. The growth of the minimum is relatively slow, from 230.332 billion yuan in 2014 to 262.48 billion yuan in 2017, but the overall trend of the average was still growing slowly, from 2277.8 billion yuan in 2014 to 289.94 billion yuan in 2017, indicating China’s production capacity has been getting stronger and stronger, and the economic strength of the provinces has been gradually improving.

The average change of CO_2_ emissions is relatively flat, but overall increases. There has been some volatility in the maximum, with moderate growth from 2014 to 2016, with a maximum of 125,863.7188 tons in 2017 below the 2016 maximum of 129,077.9661 tons, which is closely linked to energy consumption.

In terms of COD emissions and chromium emissions, they fell slowly from 2014 to 2015. The average of the two indicators in 2016 was 347,900 tons and 1762.44 kg, it is about 1/2 of 740,200 tons and 3509.51 kg of the average of the two targets in 2015, showing a sharp decline. COD emissions continued to have a sharp reduction effect in 2017, while chromium emissions rebounded to 3334.89 kg in 2017, without the sharp drop. The maximum value of COD emissions in 2017 was 1.0009 million tons close to 964,200 tons in 2016, while the maximum value of chromium emissions in 2017 was 24,134.53 kg, which was close to the 2015 peak of 26,207.95 kg.

In the health stage, the newly added health technicians and local financial health expenditures are both increasing gradually year by year, and the maximum and minimum values of health technicians are growing slowly. The maximum value increased from 603,800 in 2014 to in 2017, there were 707,500 people, and the minimum value increased from 33,900 in 2014 to 41,700 in 2017. This has a certain relationship with the training period of technicians; and the maximum value of local financial medical expenditure increased significantly. From 2014, 77.755 billion yuan increased by 53.01 billion yuan. It can be seen that the importance attached to health care in various places is also increasing in people’s attention to health.

From the output indicators of the health stage, the health index and the population mortality rate are relatively flat and fluctuating. The average health index has increased from 85.32% in 2014 to 89.85% in 2017. The average mortality rate is from the 6.14% change in 2014 was 6.18% in 2017, and the minimum and maximum values of both were almost unchanged. It shows that the development of modern medical and health is relatively mature and sound, which satisfies people’s medical guarantee for their health needs.

### 4.3. Empirical Results Analysis

The following are four aspects: the total efficiency analysis of each province, the comparative analysis of two stages of efficiency, the sub-efficiency analysis and policy analysis of each province.

#### 4.3.1. Analysis of Total Efficiency of Each Province

The total efficiency values of the two stages of DEA from economic input to healthy output in 30 provinces and municipalities (except Tibet Autonomous region) from 2014 to 2017 show that the efficiency of Beijing, Guangdong, Hainan, Ningxia Hui Autonomous Region and Qinghai Province from 2014 to 2017 is 1, which is the best resource utilization efficiency compared to other provinces.

The total efficiency of Heilongjiang and Inner Mongolia Autonomous Region in 2014 is 1, the relative efficiency is high, and the total efficiency value of 2015–2017 is less than 1. In the last three years, the total efficiency score of Heilongjiang province decreased obviously, all of which were below 0.5, and there was a lot of room for rise. In 2015 and 2017, the total efficiency score of Inner Mongolia Autonomous region was less than 0.5, which had a lot of room for improvement, and the total efficiency score of Inner Mongolia Autonomous region rebounded in 2016.

The total efficiency value of Shanxi province in 2014 is less than 1, about 0.3, and the total efficiency scores from 2015 to 2017 are 1, which indicates that the improvement efficiency of Shanxi province from 2014 to 2015 is higher. The total efficiency scores of Shanghai from 2014 to 2016 are 1, and in 2017 is less than 1. In 2014 and 2017, the total efficiency of Xinjiang Uygur Autonomous region is 1, and the integration of resource utilization from 2015 to 2016 is below 0.4. The total efficiency of the other provinces is generally low, almost all of which are between 0.2 and 0.4.

At the same time, in terms of time-varying changes in efficiency scores, there are 9 provinces that are continuing to rise or fluctuate, including Fujian province, Henan province, Hubei province, Hunan province, Jiangsu province, Shandong province, Shanxi province, Tianjin city and Zhejiang province. Except for the five provinces where the total efficiency scores have been 1 for four years, the total efficiency scores of the remaining 16 provinces have continued to decline or fluctuate, and the provinces with the largest decline are mainly Heilongjiang province, Inner Mongolia Autonomous region, Shanghai and Xinjiang. Excluding Shanghai, the total efficiency scores of the other three have dropped from 1 to around 0.3. The details are shown in Table 4.

From a geographical perspective, the total efficiency values of the provinces in the eastern region have been relatively flat. Tianjin has increased by 0.2 from 2015 to 2016, while Shanghai has declined by 0.2 from 2016 to 2017. The rest of the period is relatively flat, but overall the total efficiency value of two stages was high. These two cities have achieved a total efficiency of 1 in the eastern region, and the total efficiency of the remaining provinces and cities is 0.2–0.4, which has a lot of room for improvement.

In the middle area, it can be seen that from 2014 to 2015, Heilongjiang province and Nei Monggol have significant decreases, about 0.8, which may be related to the special situation of the year, and then the total efficiency value of Heilongjiang is more gradual, the efficiency of Nei Monggol then rises to 0.6 in 2016 and down to 0.4 again in 2017. Jilin province’s total efficiency value is flat from 2014 to 2015, then it increases to around 0.5 in 2016 and down to around 0.3 in 2017.Shanxi province rose by about 0.8 to 1 from 2014 to 2015, and then maintained a flat trend. In other provinces in the central region, the total efficiency value is between 0.2 and 0.25, which has a large room for improvement.

In the western region, Qinghai province has been fully utilized. Xinjiang has a ladder shape. After reaching its peak in 2015 and 2016, it dropped to around 0.4. The rest of the provinces remained at a low level of around 0.3 and remained flat, which has a large room for improvement. As shown in Figure 2.

#### 4.3.2. Two-Stage Efficiency Ratio

Observe the efficiency scores of the first phase and the second phase of each province from 2014 to 2017. The results are as follows:

Comparison of the first and second phases of each province in the same year

The first phase and the second phase of Beijing, Guangdong, Hainan, Ningxia Hui Autonomous region and Qinghai province are all 1, indicating that these provinces have higher production efficiency and health efficiency and better utilization of resources.

The first phase of the values of Anhui province, Fujian province, Guangxi Zhuang Autonomous region, Henan province, Hubei province, Hunan province, Jiangsu province, Liaoning province, Shandong province, Shaanxi province, Sichuan province, Yunnan province, Zhejiang province and Chongqing Municipality are larger than the values of the second stage. Comparing the total efficiency values, these show that these areas have high production efficiency and relatively good economic input, but they neglect the impact of healthy output. The first phase contributed to the overall efficiency value and the second phase dragged down the overall efficiency.

The value of the first phase of Gansu province in 2012 was 0.2159, which was smaller than the value of the second phase of 0.3520. The same was true for the next three years. Compared with the total efficiency, this showed that the healthy output of the region is better, but the output efficiency of production is relatively lower. The first phase is dragged down by overall efficiency, which may be related to the small local population and the inherent ecological environment.

There is no significant difference between the value of the first stage and the model value of the second stage in Guizhou province, at around 0.02. The value of the first stage of Jilin province from 2014 to 2015 is similar to the value of the second stage, reflecting the fact that the two provinces paid more attention to human health while developing their economies at this stage. However, the value of 0.3068 in the first phase of Jilin province in 2016 is about 3/7 of the second phase value of 0.6983. The value of the first phase in 2017 is about 0.08 less than the value of the second phase of 0.3771. Both are less than 1, and there is still room for improvement in further resource utilization.

Comparison of the first phase of different years in each province

As can be seen from the above table, the efficiency values of the first stage of each year in the provinces with efficiency values less than 1 are less than 0.5, and the efficiency values of the second stage are almost all less than 0.5, indicating that the two stages of these provinces improved. The space is large. The first stage model value of the province with a staged efficiency value less than 1 has little fluctuation from 2014 to 2017, indicating that the improvement of production efficiency is slow, which is more difficult to link with the inherent economic and resources of each region. Therefore, it is difficult to grow rapidly in the short term.

In terms of the trend of production efficiency, Guizhou province and Shanxi province have continued to increase their efficiency scores in the four-year production stage. Among them, Shanxi province has risen sharply from around 0.3 in 2014 to 1 in 2015.Most of the other provinces are in a downward trend, but the decline is not large. Among them, the greater decline was in Heilongjiang province and Inner Mongolia Autonomous region, which fell by 0.7 from 1, and the Xinjiang first rose and then fell. From 2014 to 2015, it rose by 0.7 to 1 and 2016 to 2017 decreased by 0.7.

Comparison of the second phase of different provinces in different years

The model value of the second stage of the province with a staged efficiency value less than 1 is also less volatile, but the overall fluctuation ratio is smaller than the one-stage fluctuation, reflecting that the growth of health efficiency is slower than the growth of production efficiency. It is related to the degree of thought of people’s health understanding. As shown in Table 5.

#### 4.3.3. Efficiency of Each Province

It can be seen that the efficiency of all indicators in Beijing, Guangdong, Hainan, Ningxia Hui Autonomous region and Qinghai province are 1, and all resources have been fully and effectively utilized.

The energy consumption efficiency and water efficiency of the remaining provinces are relatively flat in 2014–2017, and almost all provinces have changed by no more than 0.04 in each year, indicating that the utilization and control of energy in each region is better. Most provinces are maintained at around 50% above, a small number of provinces such as Xinjiang Autonomous region and Yunnan province are less efficient, and corresponding measures need to be taken to improve the utilization of resources. From the perspective of improving space, most provinces are less than 50%, and about 1/3 of the provinces are close to 50%, so the utilization efficiency of resources should be fine-tuned.

The efficiency of chromium emissions and CO_2_ emissions in other provinces is relatively low, below 0.4 and around, indicating that the utilization efficiency of the two is low, the emissions are high, and each region needs to control their emissions. Combined with the current situation of energy utilization, CO_2_ emissions are affected by energy utilization, but most of the energy utilization efficiency values are around 50%, indicating that the current level of CO_2_ emission control technology in China needs to be further improved. From the perspective of improving space, both are above 50% and need to be adjusted significantly.

The COD emission efficiency values of other provinces are also low. The efficiency values of almost all provinces are below 0.4 around, indicating that there is much room for improvement in the control of COD emissions in various provinces, and it is necessary to vigorously rectify. In terms of time changes, the efficiency values of COD emissions in most of the remaining provinces are declining, indicating that most provinces in China have not realized the urgency of remediation of COD emissions in recent years, and the improvement effect is poor.

The efficiency of the health technicians in the remaining provinces and the efficiency of local financial input are both low, and the improvement of the space value is greater (more than 50%), and it is necessary to increase investment in them.

The rate of population mortality is relatively high. Almost all provinces are above 0.7. The improvement space is small and negligible. It also shows that modern people have a longer life span and pay more attention to health. The simple reflection reflects people’s lives. In terms of temporal change, the remaining provinces were experiencing a decrease in the remaining index efficiency in most provinces from 2014 to 2015, excluding water efficiency and population mortality efficiency. The efficiency of the remaining indicators in most provinces decreased. From 2015 to 2016, in most provinces, the efficiency of the remaining indicators are increasing, excluding the health technicians’ efficiency. From 2016 to 2017, the efficiency of the remaining indicators in most provinces is declining, excluding the efficiency of the number of health technicians and local medical expenditures, as shown in Table 6, Table 7, Table 8 and Table 9.

### 4.4. Policy Analysis

#### 4.4.1. Energy Efficiency and CO_2_ Efficiency

China’s Energy Policy shows that China is the largest developing country in the world and faces the arduous task of developing the economy, improving people’s livelihood, and building a well-off society. Maintaining long-term, stable and sustainable use of energy resources is an important strategic task of the Chinese government. China’s energy must take a development path of high technology content, low resource consumption, low environmental pollution, good economic returns, and safe and secure development, and achieve comprehensive, clean and safe development. The basic content of China’s energy policy is: adhere to the energy development policy of “saving priority, basing domestic, diversified development, protecting the environment, technological innovation, deepening reform, international cooperation, and improving people’s livelihood”.

The National Climate Change Plan (2014–2020) proposes that by 2020, the carbon dioxide emissions per unit of GDP will be 40%–45% lower than that of 2005, and non-fossil energy will account for about 15% of primary energy consumption. The area of forests and the volume of reserves increased by 40 million hectares and 1.3 billion cubic meters respectively compared with 2005.

It can be seen from Figure 2 that the energy efficiency is always higher than or equal to the efficiency of CO_2_ emissions. Except for Beijing, Guangdong, Hainan, Ningxia Hui Autonomous region and Qinghai province, the efficiency values of the other provinces are quite different.

According to the analysis in Table 5, the CO_2_ emission efficiency value is low, but there is a tendency to increase slowly. Combined with the current status of energy utilization, CO_2_ emissions are affected by energy utilization, but most of the energy utilization efficiency values are about 50%, indicating that the current level of China’s CO_2_ emission control technology needs to be further improved, as shown in Figure 3.

#### 4.4.2. Water Efficiency and Chemical Oxygen Demand (COD) Efficiency, and Chromium Emission Efficiency

Water use efficiency is relatively flat in 2014–2017, and most provinces are maintained at around 50%, which is related to national water-saving control, but a few provinces such as Xinjiang and Yunnan Province are less efficient, which is related to local government-related policies. It is related to the local population and geographical environment.

For COD emission efficiency and chromium emission efficiency, the utilization efficiency of both are low, and the emissions are high, which is closely related to the development of the national economy.

Except for the efficiency values of all three are 1 of Beijing, Guangdong, Hainan, Ningxia and Qinghai province, water efficiency in Hebei, Henan, Shanxi, Shandong, Shaanxi and Sichuan provinces are higher than that in chromium and COD. The water efficiency values of the other provinces are not the highest, indicating that these provinces can focus on improving the water efficiency value to further improve the efficiency values of the other two, as shown in Figure 4.

#### 4.4.3. Government Health Expenditure Efficiency and Population Mortality Efficiency Value

As can be seen from the Figure 5, the efficiency value of local fiscal expenditure is less than or equal to the efficiency value of population death, and the difference between the two is large. But over the years, the gap between the two provinces becomes smaller. Local fiscal expenditure efficiency is low in all provinces and needs to increase investment in them. However, the overall growth is also slow, which is directly related to the “Opinions on Integrating the Basic Medical Insurance System for Urban and Rural Residents” issued by the State Council in 2016.

## 5. Policy Suggestion

This paper uses the modified undesirable dynamic network model to evaluate the economic input and health output productivity and health efficiency of 30 provinces (excluding Tibet Autonomous Region). The main conclusions are as follows:

Policy measures for comprehensive governance should be formulated. From the above conclusions, in terms of energy efficiency and CO_2_ emission efficiency, energy efficiency is always higher than or equal to CO_2_ emission efficiency. Except Beijing, Guangdong, Hainan, Ningxia Hui Autonomous region, Qinghai province, other provinces have large differences in efficiency values and low CO_2_ emission efficiency values. Compared with water use efficiency, COD emission efficiency and chromium emission efficiency, most provinces have lower water efficiency values than chromium emissions efficiency value or COD emission efficiency value. This indicates that the current CO_2_ emission efficiency and water efficiency values of all provinces in China are low, and there is much room for improvement. This should start at the administration and focus on the integrated design and implementation of carbon dioxide governance and water cleansing initiatives. According to the specific conditions of each province, policy initiatives for the treatment of carbon dioxide and water cleanliness are implemented together and arranged in an overall manner, thereby carrying out governance more efficiently.

From the conclusion, we know that the efficiency of the employment population in each province is low, and there is large room for improvement. Comparing local fiscal expenditure and population death index, the efficiency value of local fiscal expenditure is less than or equal to the efficiency value of population death, and the difference between the two is large. For the former, there are widespread employment mismatches, unreasonable structures, frictional unemployment and structural unemployment in various provinces. Each province needs to pay attention to the adjustment of industrial structure in the region, rationally cultivate talents, match industry counterparts, and improve each the comprehensive quality of talents is aimed at cultivating compound talents. For the latter, we should ensure that the medical investment can solve the problem of human health from the aspects of system design, medical guarantee, technical improvement and technical support, the retention of medical personnel, the improvement of medical conditions and so on.

Starting from education, expanding the influence of health and efficiency guarantees, and building a sense of economic input from the next generation to ensure the healthy output will thus help the next generation to develop a healthy and environmentally friendly society. In the enterprise aspect, the human body and the sense of well-being protection should be added to the education of entrepreneurs, thus fundamentally affecting their decision-making, further improving the efficiency of carbon dioxide emissions and the efficiency of water cleaning, and playing a positive feedback role on the output of health efficiency.

Increase the intensity of scientific propaganda, strengthening the scientific understanding of health among adolescents and residents, and establishing a sense of crisis awareness and self-protection is also necessary. Actively adopting corresponding measures, and regulating and control in the process of economic input to make healthy output has a more obvious effect. Local governments and schools should increase their explanation and publicity on the source of health efficiency, so that residents can realize that while increasing production efficiency, it is necessary to ensure the excellent results of health output. In order to ensure their health in order to develop local gross domestic product, it is necessary to adhere to the requirements of sustainable development.

## 6. Conclusions

From the evaluation results of 30 provinces, except for the efficiency values of all indicators in Beijing, Guangdong, Hainan, Ningxia and Qinghai province, in which all resources are fully utilized, the efficiency values of the first stage are greater than the efficiency values of the second stage in the most of the remaining provinces, and the efficiency values of the second stage plays a drag on the total efficiency value. However, overall, production efficiency and health efficiency show a slight growth trend year by year, and the fluctuation of health efficiency is less than the growth fluctuation of production efficiency. This shows that the efficiency of production input utilization is high in all provinces of our country, but the reflection of human health efficiency is not optimistic.

Except for Beijing, Guangdong, Hainan, Ningxia and Qinghai province, the total efficiency values of other provinces are less than 1, and the fluctuations from 2014 to 2017 are not large, and most provinces are between 0.2 and 0.4, indicating that the overall efficiency value of China’s economic input to healthy output is generally low, and should focus on greatly improving health efficiency. The total efficiency scores showed that there were only 9 provinces that continued to rise or fluctuate, and the largest increase was in Shanxi province and Xinjiang. The total efficiency scores of the remaining 16 provinces continued to decline or fluctuate. The largest declines including Heilongjiang province, Inner Mongolia and Shanghai.

From the perspective of various indicators, the employment population efficiency, total water use efficiency, COD efficiency, CO_2_ emission efficiency, chromium emission efficiency, health technician efficiency and local financial expenditure efficiency are lower, the space of improvement is large and need to be adjusted significantly. The efficiency of COD efficiency, CO_2_ emissions, and chromium emissions are related to energy consumption and water consumption. Therefore, the focus can be on energy and water consumption and medical input.

In terms of intermediate indicators, the GDP efficiency value is the highest in most provinces. Gansu province, Guizhou province, Shanxi province, Xinjiang have lower adjusted GDP efficiency values, and the improvement of spatial value is large. This is related to the local geographical environment and climate. The efficiency of health technicians and the efficiency of local financial inputs are both low, and investment in them needs to be increased. The efficiency of health care in developed regions are relatively high.

For signs of global warming, we need to control emissions of related gases. The efficiency of CO_2_ emissions in various regions is relatively low, indicating that the control technology for CO_2_ in various regions of China needs to be improved. From the perspective of energy, we can improve the technical level of clean energy utilization, take the replacement of new energy sources and promote the use of clean energy.

Comparing energy efficiency and CO_2_ emission efficiency, energy efficiency is always higher than or equal to CO_2_ emission efficiency. Except for Beijing, Guangdong, Hainan, Ningxia, Qinghai province, the difference in other provinces are large. The CO_2_ emission efficiency value is low, but there is a tendency to increase slowly. Combined with the current situation of energy utilization, CO_2_ emissions are affected by energy utilization, but the energy utilization efficiency value is mostly around 50%, indicating that China’s current CO_2_ the level of emission control technology needs to be further improved.

Comparing water-use efficiency, COD emission efficiency and chromium emission efficiency, except for Beijing, Guangdong, Hainan, Ningxia and Qinghai province, the efficiency values of all three are 1. Hebei, Henan, Shanxi, Shandong province, Shaanxi province and Sichuan Province, the water efficiency values are higher than the emission efficiency values of chromium and the emission efficiency value of COD. The water efficiency values of other provinces are not the highest, indicating that these provinces can focus on improving the water efficiency value to further improve The efficiency values of the remaining two.

## Figures and Tables

**Figure 1 ijerph-16-04827-f001:**
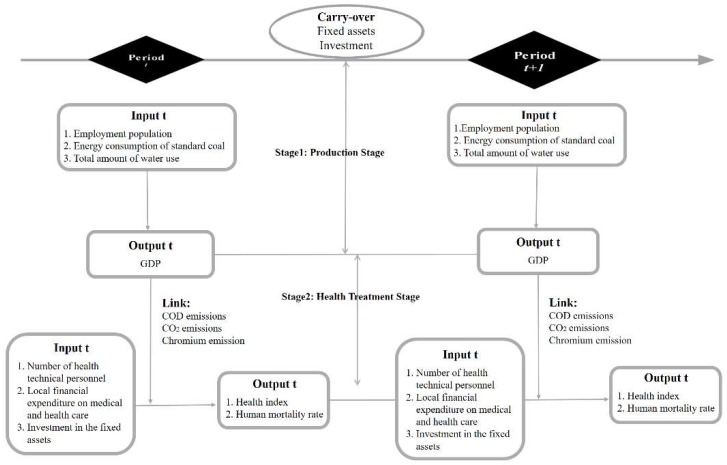
Network dynamic model.

**Figure 2 ijerph-16-04827-f002:**
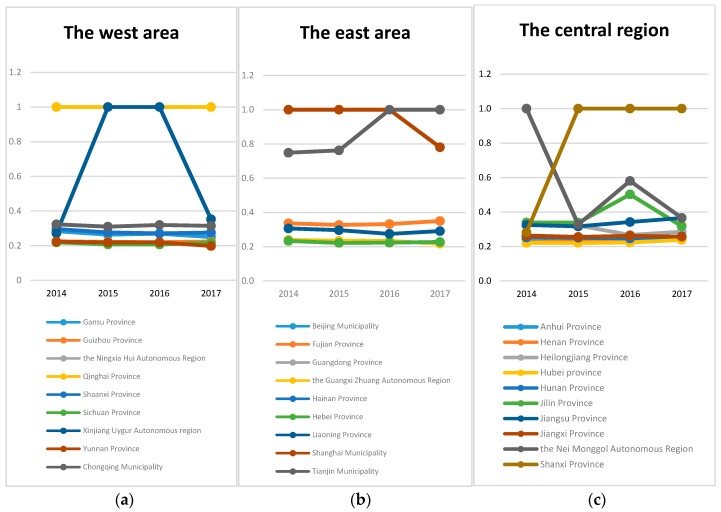
Total efficiency of 30 provinces from 2014–2017: (**a**) description of total efficiency of the central provinces; (**b**) Description of total efficiency of the eastern provinces; (**c**) description of total efficiency of the western provinces.

**Figure 3 ijerph-16-04827-f003:**
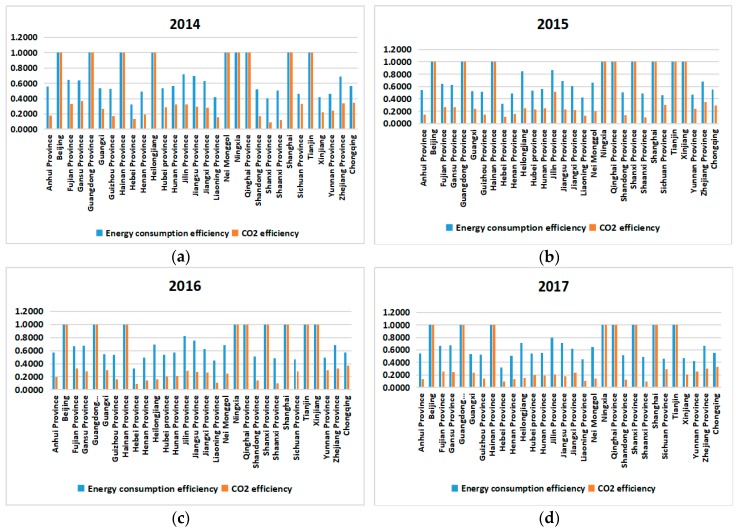
Comparison of energy consumption standard coal efficiency and CO_2_ emission efficiency, (**a**) 2014, (**b**) 2015, (**c**) 2016, (**d**) 2017.

**Figure 4 ijerph-16-04827-f004:**
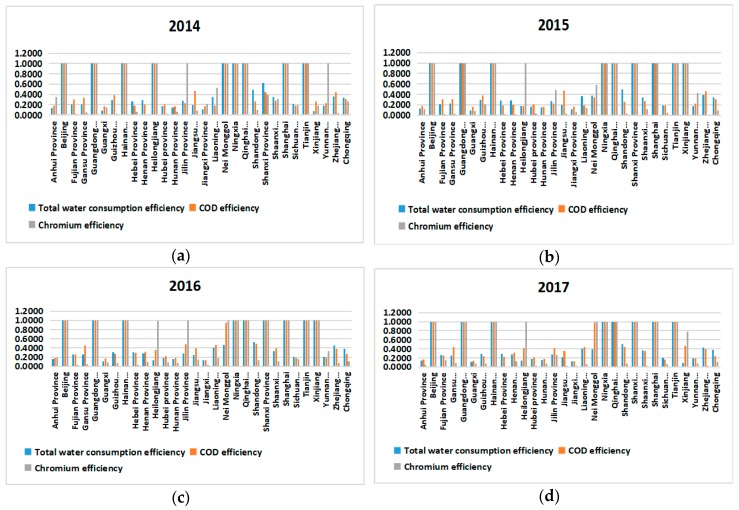
Comparison of total water use efficiency, chemical oxygen demand (COD) efficiency, chromium emission efficiency and total water consumption efficiency (**a**) 2014, (**b**) 2015, (**c**) 2016, (**d**) 2017.

**Figure 5 ijerph-16-04827-f005:**
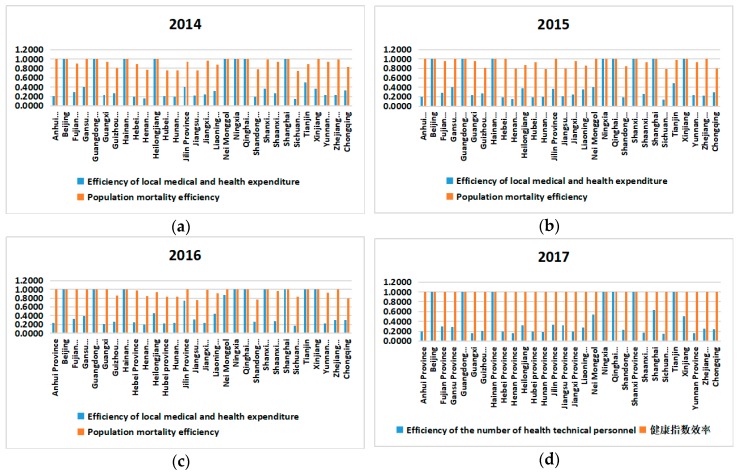
Comparison of the efficiency of government health expenditure and the efficiency of population mortality, (**a**) 2014, (**b**) 2015, (**c**) 2016, (**d**) 2017.

**Table 1 ijerph-16-04827-t001:** Input and output variables.

Stage	Input	Output
Production Stage	Employment population Energy consumption of standard coal Total amount of water use Investment in the fixed assets	GDP COD emissions CO_2_ emissions Chromium emission Health index Human mortality rate



Health Stage	GDP
	COD emissions
	CO_2_ emissions	
Chromium emission	
Number of health technical personnel	
Local financial expenditure on medical and health care	
Investment in the fixed assets	

**Table 2 ijerph-16-04827-t002:** Standard conversion factor and carbon emission coefficient for various types of energy.

Coefficient	Raw Coal	Coke	Crude Oil	Gasoline	Diesel Oil	Kerosene	Fuel Oil	Natural Gas	Power
Standard conversion coefficient	0.71	0.97	1.43	1.47	1.46	1.47	1.43	13.3	1.23
Carbon emission coefficient	0.75	0.11	0.59	0.55	0.59	0.34	0.62	0.45	2.21

**Table 3 ijerph-16-04827-t003:** Descriptive statistics of variables.

Year	Variable	Average	Maximum	Minimum	Standard
2014	Number of employed population (10,000)	351.88	1012.02	44.55	252.40
	Energy consumption standard coal (10,000 tons)	14,664.84	36,511.00	1819.93	8392.43
	Total water use (100 million cubic meters)	202.15	591.29	24.09	146.08
	Fixed asset investment stock (100 million yuan)	73,291.67	191,337.95	10,690.67	46,804.02
	GDP (100 million yuan)	22,780.95	67,809.85	2303.32	16,527.77
	COD emissions (10,000 tons)	76.39	178.04	10.50	46.29
	CO_2_ emissions (tons)	40,254.90	119,050.41	4949.07	27,841.69
	Chromium emissions (kg)	4426.51	27,844.00	12.62	6223.18
	Number of health technicians (10,000)	25.22	60.38	3.39	14.80
	Local financial and health expenditure (100 million yuan)	334.59	777.55	65.27	166.01
	Health index (%)	85.32	90.20	78.64	2.65
	Population mortality rate (%)	6.14	7.18	4.53	0.71
2015	Number of employed population (10,000)	388.19	1153.64	46.07	281.24
	Energy consumption standard coal (10,000 tons)	14,910.60	37,945.40	1937.77	8535.14
	Total water use (100 million cubic meters)	202.43	577.20	25.70	144.28
	Fixed asset investment stock (100 million yuan)	84,773.24	221,281.95	12,874.99	53,780.07
	GDP (100 million yuan)	24,058.05	72,812.55	2417.05	17,743.14
	COD emissions (10,000 tons)	74.02	175.76	10.43	45.17
	CO_2_ emissions (tons)	39,949.16	124,607.70	4228.68	28,554.09
	Chromium emissions (kg)	3509.51	26,207.95	6.96	5520.20
	Number of health technicians (10,000)	26.61	61.82	3.54	15.56
	Local financial and health expenditure (100 million yuan)	393.53	918.36	74.11	195.15
	Health index (%)	85.02	90.25	78.30	2.78
	Population mortality rate (%)	6.01	7.20	4.32	0.79
2016	Number of employed population (10,000)	427.33	1281.20	52.10	307.89
	Energy consumption standard coal (10,000 tons)	15,192.23	38,722.80	2006.12	8711.75
	Total water use (100 million cubic meters)	200.31	577.40	26.40	142.67
	Fixed asset investment stock (100 million yuan)	96,525.68	253,361.82	15,167.04	61,190.82
	GDP (100 million yuan)	25,963.95	80,854.91	2572.49	19,602.87
	COD emissions (10,000 tons)	34.79	96.42	7.03	21.79
	CO_2_ emissions (tons)	40,060.58	129,077.97	4690.33	29,080.87
	Chromium emissions (kg)	1762.44	7411.17	5.96	2149.43
	Number of health technicians (10,000)	28.10	66.53	3.70	16.45
	Local financial and health expenditure (100 million yuan)	433.25	1121.83	82.03	227.30
	Health index (%)	83.93	89.62	77.69	2.79
	Population mortality rate (%)	6.10	7.24	4.26	0.81
2017	Number of employed population (10,000)	472.68	1426.13	59.42	341.88
	Energy consumption standard coal (10,000 tons)	15,610.47	38,683.70	2103.13	8773.60
	Total water use (100 million cubic meters)	200.40	591.30	25.80	142.73
	Fixed asset investment stock (100 million yuan)	108,217.49	284,241.81	17,594.56	69,468.26
	GDP (100 million yuan)	28,194.31	89,705.23	2624.83	21,655.15
	COD emissions (10,000 tons)	33.98	100.09	5.75	22.44
	CO_2_ emissions (tons)	41,145.89	125,863.72	4013.87	30,008.59
	Chromium emissions (kg)	3334.89	24,134.53	52.08	5248.49
	Number of health technicians (10,000)	29.87	70.75	4.17	17.55
	Local financial and health expenditure (100 million yuan)	474.98	1307.56	97.98	256.34
	Health index (%)	82.87	89.85	77.58	3.05
	Population mortality rate (%)	6.18	7.40	4.48	0.81

**Table 4 ijerph-16-04827-t004:** Overall efficiency by provinces from 2014 to 2017.

No.	DMU	2014	2015	2016	2017
1	Anhui Province	0.2453	0.2370	0.2478	0.2394
2	Beijing Municipality	1.0000	1.0000	1.0000	1.0000
3	Fujian Province	0.3363	0.3269	0.3321	0.3500
4	Gansu Province	0.2839	0.2635	0.2695	0.2499
5	Guangdong Province	1.0000	1.0000	1.0000	1.0000
6	the Guangxi Zhuang Autonomous Region	0.2393	0.2343	0.2334	0.2174
7	Guizhou Province	0.2250	0.2218	0.2205	0.2236
8	Hainan Province	1.0000	1.0000	1.0000	1.0000
9	Hebei Province	0.2336	0.2218	0.2235	0.2273
10	Henan Province	0.2485	0.2378	0.2425	0.2505
11	Heilongjiang Province	1.0000	0.3217	0.2674	0.2862
12	Hubei province	0.2215	0.2206	0.2237	0.2384
13	Hunan Province	0.2522	0.2502	0.2471	0.2593
14	Jilin Province	0.3395	0.3388	0.5026	0.3181
15	Jiangsu Province	0.3249	0.3166	0.3422	0.3656
16	Jiangxi Province	0.2633	0.2553	0.2619	0.2572
17	Liaoning Province	0.3065	0.2962	0.2749	0.2911
18	the Nei Monggol Autonomous Region	1.0000	0.3279	0.5804	0.3663
19	the Ningxia Hui Autonomous Region	1.0000	1.0000	1.0000	1.0000
20	Qinghai Province	1.0000	1.0000	1.0000	1.0000
21	Shandong Province	0.3016	0.2906	0.3166	0.3278
22	Shanxi Province	0.2819	1.0000	1.0000	1.0000
23	Shaanxi Province	0.2963	0.2766	0.2708	0.2759
24	Shanghai Municipality	1.0000	1.0000	1.0000	0.7804
25	Sichuan Province	0.2189	0.2076	0.2070	0.2168
26	Tianjin Municipality	0.7488	0.7623	1.0000	1.0000
27	Xinjiang Uygur Autonomous region	0.2720	1.0000	1.0000	0.3527
28	Yunnan Province	0.2231	0.2185	0.2190	0.1983
29	Zhejiang Province	0.3285	0.3240	0.3516	0.3726
30	Chongqing Municipality	0.3232	0.3100	0.3197	0.3145

**Table 5 ijerph-16-04827-t005:** Comparison of two-stage efficiency scores from 2014–2017.

DMU	2014 Total	2014 (I)	2014 (II)	2015 Total	2015 (I)	2015 (II)	2016 Total	2016 (I)	2016 (II)	2017 Total	2017 (I)	2017 (II)
Anhui	0.2453	0.3059	0.1847	0.2370	0.2872	0.1869	0.2478	0.2988	0.1968	0.2394	0.2734	0.2054
Beijing	1.0000	1.0000	1.0000	1.0000	1.0000	1.0000	1.0000	1.0000	1.0000	1.0000	1.0000	1.0000
Fujian	0.3363	0.4207	0.2520	0.3269	0.3957	0.2582	0.3321	0.3927	0.2716	0.3500	0.3819	0.3181
Gansu	0.2839	0.2159	0.3520	0.2635	0.1837	0.3433	0.2695	0.1991	0.3399	0.2499	0.1515	0.3483
Guangdong	1.0000	1.0000	1.0000	1.0000	1.0000	1.0000	1.0000	1.0000	1.0000	1.0000	1.0000	1.0000
Guangxi	0.2393	0.2954	0.1833	0.2343	0.2847	0.1839	0.2334	0.2941	0.1726	0.2174	0.2452	0.1896
Guizhou	0.2250	0.2264	0.2237	0.2218	0.2267	0.2168	0.2205	0.2327	0.2082	0.2236	0.2268	0.2205
Hainan	1.0000	1.0000	1.0000	1.0000	1.0000	1.0000	1.0000	1.0000	1.0000	1.0000	1.0000	1.0000
Hebei	0.2336	0.2980	0.1692	0.2218	0.2689	0.1747	0.2235	0.2492	0.1977	0.2273	0.2367	0.2179
Henan	0.2485	0.3727	0.1243	0.2378	0.3526	0.1230	0.2425	0.3373	0.1476	0.2505	0.3244	0.1766
Heilongjiang	1.0000	1.0000	1.0000	0.3217	0.3294	0.3140	0.2674	0.2020	0.3328	0.2862	0.1837	0.3888
Hubei	0.2215	0.2813	0.1617	0.2206	0.2790	0.1622	0.2237	0.2820	0.1653	0.2384	0.2798	0.1970
Hunan	0.2522	0.3491	0.1554	0.2502	0.3466	0.1538	0.2471	0.3238	0.1704	0.2593	0.3222	0.1964
Jilin	0.3395	0.3423	0.3366	0.3388	0.3697	0.3078	0.5026	0.3068	0.6983	0.3181	0.2591	0.3771
Jiangsu	0.3249	0.4602	0.1895	0.3166	0.4478	0.1855	0.3422	0.4457	0.2388	0.3656	0.4038	0.3273
Jiangxi	0.2633	0.3107	0.2159	0.2553	0.2973	0.2134	0.2619	0.3163	0.2075	0.2572	0.2939	0.2204
Liaoning	0.3065	0.3623	0.2507	0.2962	0.3225	0.2698	0.2749	0.2416	0.3081	0.2911	0.2342	0.3480
Nei Monggol	1.0000	1.0000	1.0000	0.3279	0.2667	0.3892	0.5804	0.2926	0.8683	0.3663	0.1793	0.5534
Ningxia	1.0000	1.0000	1.0000	1.0000	1.0000	1.0000	1.0000	1.0000	1.0000	1.0000	1.0000	1.0000
Qinghai	1.0000	1.0000	1.0000	1.0000	1.0000	1.0000	1.0000	1.0000	1.0000	1.0000	1.0000	1.0000
Shandong	0.3016	0.4511	0.1522	0.2906	0.4268	0.1544	0.3166	0.4445	0.1887	0.3278	0.4053	0.2504
Shanxi	0.2819	0.2630	0.3007	1.0000	1.0000	1.0000	1.0000	1.0000	1.0000	1.0000	1.0000	1.0000
Shaanxi	0.2963	0.3852	0.2074	0.2766	0.3488	0.2043	0.2708	0.3413	0.2004	0.2759	0.3361	0.2157
Shanghai	1.0000	1.0000	1.0000	1.0000	1.0000	1.0000	1.0000	1.0000	1.0000	0.7804	1.0000	0.5609
Sichuan	0.2189	0.3218	0.1160	0.2076	0.2978	0.1175	0.2070	0.2829	0.1310	0.2168	0.2811	0.1526
Tianjin	0.7488	1.0000	0.4976	0.7623	1.0000	0.5245	1.0000	1.0000	1.0000	1.0000	1.0000	1.0000
Xinjiang	0.2720	0.2236	0.3203	1.0000	1.0000	1.0000	1.0000	1.0000	1.0000	0.3527	0.1443	0.5612
Yunnan	0.2231	0.2432	0.2030	0.2185	0.2432	0.1939	0.2190	0.2566	0.1815	0.1983	0.2174	0.1792
Zhejiang	0.3285	0.4540	0.2030	0.3240	0.4521	0.1959	0.3516	0.4599	0.2432	0.3726	0.4390	0.3062
Chongqing	0.3232	0.3851	0.2613	0.3100	0.3813	0.2387	0.3197	0.4091	0.2303	0.3145	0.3780	0.2511

**Table 6 ijerph-16-04827-t006:** Efficiency values of inter-provincial key indicators for 2014.

DMU	Energy	Water	COD	CO_2_	Chromium	Health Tech	Health Exp	Mortality
Anhui	0.5559	0.1347	0.1863	0.1757	0.3439	0.1673	0.2022	0.9994
Beijing	1.0000	1.0000	1.0000	1.0000	1.0000	1.0000	1.0000	1.0000
Fujian	0.6443	0.2051	0.3068	0.3344	0.0293	0.2356	0.2928	0.9030
Gansu	0.6361	0.2099	0.3417	0.3659	0.0506	0.2987	0.4052	1.0000
Guangdong	1.0000	1.0000	1.0000	1.0000	1.0000	1.0000	1.0000	1.0000
Guangxi	0.5330	0.0905	0.1685	0.2658	0.1463	0.1471	0.2293	0.9461
Guizhou	0.5275	0.2949	0.3874	0.1710	0.0214	0.2238	0.2664	0.8083
Hainan	1.0000	1.0000	1.0000	1.0000	1.0000	1.0000	1.0000	1.0000
Hebei	0.3214	0.2682	0.1835	0.1323	0.0651	0.1552	0.2015	0.8916
Henan	0.4888	0.2934	0.2097	0.1941	0.0157	0.1212	0.1560	0.7694
Heilongjiang	1.0000	1.0000	1.0000	1.0000	1.0000	1.0000	1.0000	1.0000
Hubei	0.5374	0.1669	0.2098	0.2875	0.0253	0.1523	0.2109	0.7538
Hunan	0.5653	0.1430	0.1741	0.3211	0.0571	0.1485	0.1994	0.7607
Jilin	0.7171	0.2760	0.2239	0.3267	1.0000	0.2861	0.4067	0.9419
Jiangsu	0.6980	0.1935	0.4682	0.2951	0.0841	0.2023	0.2217	0.7625
Jiangxi	0.6342	0.1081	0.1753	0.2792	0.2162	0.1921	0.2470	0.9665
Liaoning	0.4205	0.3549	0.1861	0.1593	0.5226	0.2083	0.3232	0.8796
Nei Monggol	1.0000	1.0000	1.0000	1.0000	1.0000	1.0000	1.0000	1.0000
Ningxia	1.0000	1.0000	1.0000	1.0000	1.0000	1.0000	1.0000	1.0000
Qinghai	1.0000	1.0000	1.0000	1.0000	1.0000	1.0000	1.0000	1.0000
Shandong	0.5213	0.4869	0.2641	0.1724	0.0945	0.1431	0.1951	0.7772
Shanxi	0.4045	0.6178	0.4498	0.0936	0.3909	0.2419	0.3641	0.9852
Shaanxi	0.5048	0.3462	0.2773	0.1184	0.3177	0.1608	0.2660	0.9426
Shanghai	1.0000	1.0000	1.0000	1.0000	1.0000	1.0000	1.0000	1.0000
Sichuan	0.4597	0.2117	0.1857	0.3295	0.1992	0.1156	0.1463	0.7428
Tianjin	1.0000	1.0000	1.0000	1.0000	1.0000	0.5500	0.5007	0.8883
Xinjiang	0.4224	0.0748	0.2624	0.2204	0.1774	0.2743	0.3662	1.0000
Yunnan	0.4649	0.1782	0.2250	0.2462	1.0000	0.1807	0.2366	0.9441
Zhejiang	0.6866	0.3657	0.4407	0.3379	0.0391	0.1760	0.2313	0.9940
Chongqing	0.5679	0.3328	0.3121	0.3469	0.2698	0.2406	0.3269	0.8280

**Table 7 ijerph-16-04827-t007:** Efficiency values of inter-provincial key indicators for 2015.

DMU	Energy	Water	COD	CO_2_	Chromium	Health Tech	Health Exp	Mortality
Anhui	0.5444	0.1295	0.1816	0.1455	0.1195	0.1688	0.2049	1.0000
Beijing	1.0000	1.0000	1.0000	1.0000	1.0000	1.0000	1.0000	1.0000
Fujian	0.6403	0.2160	0.3016	0.2700	0.0107	0.2483	0.2806	0.9519
Gansu	0.6291	0.2217	0.3088	0.2620	0.0381	0.2851	0.4015	1.0000
Guangdong	1.0000	1.0000	1.0000	1.0000	1.0000	1.0000	1.0000	1.0000
Guangxi	0.5224	0.0942	0.1692	0.2390	0.0817	0.1419	0.2342	0.9550
Guizhou	0.5172	0.2926	0.3813	0.1425	0.2135	0.2088	0.2664	0.8083
Hainan	1.0000	1.0000	1.0000	1.0000	1.0000	1.0000	1.0000	1.0000
Hebei	0.3222	0.2817	0.1847	0.1094	0.0201	0.1564	0.1930	1.0000
Henan	0.4846	0.2808	0.2055	0.1591	0.0058	0.1212	0.1495	0.7990
Heilongjiang	0.8515	0.1719	0.1793	0.2443	1.0000	0.2831	0.3850	0.8725
Hubei	0.5364	0.1628	0.2103	0.2327	0.0356	0.1481	0.1876	0.9309
Hunan	0.5615	0.1465	0.1695	0.2506	0.0145	0.1457	0.1945	0.7882
Jilin	0.8676	0.2680	0.2225	0.5162	0.4841	0.2499	0.3657	1.0000
Jiangsu	0.6916	0.2029	0.4673	0.2317	0.0288	0.1913	0.2156	0.8061
Jiangxi	0.6094	0.1164	0.1693	0.2224	0.0570	0.1868	0.2481	0.9622
Liaoning	0.4244	0.3641	0.1856	0.1310	0.1441	0.2160	0.3598	0.8660
Nei Monggol	0.6665	0.3831	0.3418	0.2045	0.5794	0.3771	0.4012	1.0000
Ningxia	1.0000	1.0000	1.0000	1.0000	1.0000	1.0000	1.0000	1.0000
Qinghai	1.0000	1.0000	1.0000	1.0000	1.0000	1.0000	1.0000	1.0000
Shandong	0.5031	0.5001	0.2560	0.1321	0.0384	0.1415	0.1907	0.8478
Shanxi	1.0000	1.0000	1.0000	1.0000	1.0000	1.0000	1.0000	1.0000
Shaanxi	0.4851	0.3474	0.2738	0.0958	0.1107	0.1584	0.2636	0.9342
Shanghai	1.0000	1.0000	1.0000	1.0000	1.0000	1.0000	1.0000	1.0000
Sichuan	0.4610	0.1925	0.1821	0.3048	0.0466	0.1178	0.1427	0.7827
Tianjin	1.0000	1.0000	1.0000	1.0000	1.0000	0.5666	0.4919	0.9820
Xinjiang	1.0000	1.0000	1.0000	1.0000	1.0000	1.0000	1.0000	1.0000
Yunnan	0.4732	0.1782	0.2271	0.2353	0.4305	0.1675	0.2341	0.9281
Zhejiang	0.6848	0.3944	0.4600	0.3507	0.0146	0.1651	0.2266	1.0000
Chongqing	0.5495	0.3427	0.3052	0.2894	0.0986	0.2259	0.3004	0.7952

**Table 8 ijerph-16-04827-t008:** Efficiency values of inter-provincial key indicators for 2016.

DMU	Energy	Water	COD	CO_2_	Chromium	Health Tech	Health Exp	Mortality
Anhui	0.5705	0.1529	0.1777	0.1944	0.1990	0.1623	0.2313	1.0000
Beijing	1.0000	1.0000	1.0000	1.0000	1.0000	1.0000	1.0000	1.0000
Fujian	0.6665	0.2580	0.2593	0.3300	0.0289	0.2200	0.3232	1.0000
Gansu	0.6790	0.2624	0.4502	0.2849	0.0475	0.2928	0.3871	0.9999
Guangdong	1.0000	1.0000	1.0000	1.0000	1.0000	1.0000	1.0000	1.0000
Guangxi	0.5449	0.1063	0.1718	0.3012	0.0827	0.1297	0.2155	1.0000
Guizhou	0.5345	0.3047	0.2741	0.1598	0.0685	0.1865	0.2598	0.8570
Hainan	1.0000	1.0000	1.0000	1.0000	1.0000	1.0000	1.0000	1.0000
Hebei	0.3292	0.3084	0.2986	0.0950	0.0354	0.1450	0.2550	0.9769
Henan	0.4939	0.2857	0.3169	0.1395	0.0931	0.1200	0.1975	0.8488
Heilongjiang	0.6928	0.1382	0.3575	0.1595	0.9804	0.2286	0.4567	0.9412
Hubei	0.5324	0.1789	0.2184	0.2026	0.0835	0.1374	0.2205	0.8351
Hunan	0.5693	0.1555	0.1869	0.2129	0.0647	0.1336	0.2364	0.8284
Jilin	0.8220	0.2868	0.4754	0.2898	1.0000	0.6529	0.7437	1.0000
Jiangsu	0.7538	0.2487	0.3929	0.2734	0.1497	0.2226	0.3148	0.7521
Jiangxi	0.6261	0.1276	0.1301	0.2657	0.0363	0.1774	0.2391	0.9924
Liaoning	0.4524	0.4013	0.4614	0.1123	0.1857	0.2000	0.4447	0.9074
Nei Monggol	0.6856	0.4659	0.9499	0.2455	1.0000	0.8686	0.8679	1.0000
Ningxia	1.0000	1.0000	1.0000	1.0000	1.0000	1.0000	1.0000	1.0000
Qinghai	1.0000	1.0000	1.0000	1.0000	1.0000	1.0000	1.0000	1.0000
Shandong	0.5081	0.5283	0.4877	0.1472	0.1265	0.1607	0.2612	0.7646
Shanxi	1.0000	1.0000	1.0000	1.0000	1.0000	1.0000	1.0000	1.0000
Shaanxi	0.4848	0.3359	0.3920	0.1018	0.1110	0.1347	0.2737	0.9627
Shanghai	1.0000	1.0000	1.0000	1.0000	1.0000	1.0000	1.0000	1.0000
Sichuan	0.4691	0.2063	0.1766	0.2810	0.1529	0.1110	0.1730	0.8335
Tianjin	1.0000	1.0000	1.0000	1.0000	1.0000	1.0000	1.0000	1.0000
Xinjiang	1.0000	1.0000	1.0000	1.0000	1.0000	1.0000	1.0000	1.0000
Yunnan	0.4930	0.2050	0.1907	0.2969	0.3361	0.1550	0.2219	0.9228
Zhejiang	0.6845	0.4491	0.3797	0.3277	0.0677	0.1897	0.2968	1.0000
Chongqing	0.5682	0.3829	0.2671	0.3696	0.1048	0.2070	0.2998	0.7992

**Table 9 ijerph-16-04827-t009:** Efficiency values of inter-provincial key indicators for 2017.

DMU	Energy	Water	COD	CO_2_	Chromium	Health Tech	Health Exp	Mortality
Anhui	0.5475	0.1349	0.1647	0.1330	0.0468	0.1986	0.2160	0.9814
Beijing	1.0000	1.0000	1.0000	1.0000	1.0000	1.0000	1.0000	1.0000
Fujian	0.6690	0.2670	0.2515	0.2558	0.1457	0.3039	0.3460	0.9569
Gansu	0.6765	0.2450	0.4379	0.2442	0.0834	0.2873	0.4304	0.9393
Guangdong	1.0000	1.0000	1.0000	1.0000	1.0000	1.0000	1.0000	1.0000
Guangxi	0.5332	0.1083	0.1392	0.2371	0.0809	0.1557	0.2354	0.9378
Guizhou	0.5294	0.2921	0.2326	0.1455	0.0689	0.2114	0.2689	0.8216
Hainan	1.0000	1.0000	1.0000	1.0000	1.0000	1.0000	1.0000	1.0000
Hebei	0.3208	0.2874	0.2314	0.0947	0.0149	0.1940	0.2663	0.8879
Henan	0.5085	0.2726	0.3136	0.1323	0.1195	0.1665	0.2184	0.8205
Heilongjiang	0.7086	0.1434	0.4103	0.1550	1.0000	0.3238	0.5013	0.8774
Hubei	0.5402	0.1779	0.2077	0.1943	0.0206	0.1919	0.2463	0.7760
Hunan	0.5561	0.1505	0.1807	0.1929	0.0705	0.1834	0.2545	0.7701
Jilin	0.8004	0.2761	0.4195	0.2050	0.2666	0.3332	0.4506	0.9217
Jiangsu	0.7100	0.2162	0.3461	0.1814	0.0196	0.3196	0.3877	0.8391
Jiangxi	0.6164	0.1235	0.1222	0.2331	0.0121	0.2015	0.2465	0.9671
Liaoning	0.4497	0.4084	0.4400	0.1068	0.0603	0.2803	0.4735	0.8337
Nei Monggol	0.6488	0.3931	0.9807	0.1466	1.0000	0.5403	0.5665	1.0000
Ningxia	1.0000	1.0000	1.0000	1.0000	1.0000	1.0000	1.0000	1.0000
Qinghai	1.0000	1.0000	1.0000	1.0000	1.0000	1.0000	1.0000	1.0000
Shandong	0.5113	0.5000	0.4420	0.1197	0.0732	0.2296	0.3465	1.0000
Shanxi	1.0000	1.0000	1.0000	1.0000	1.0000	1.0000	1.0000	1.0000
Shaanxi	0.4848	0.3359	0.3920	0.1018	0.1110	0.1347	0.2737	0.9627
Shanghai	1.0000	1.0000	1.0000	1.0000	1.0000	1.0000	1.0000	1.0000
Sichuan	0.4691	0.2063	0.1766	0.2810	0.1529	0.1110	0.1730	0.8335
Tianjin	1.0000	1.0000	1.0000	1.0000	1.0000	1.0000	1.0000	1.0000
Xinjiang	1.0000	1.0000	1.0000	1.0000	1.0000	1.0000	1.0000	1.0000
Yunnan	0.4930	0.2050	0.1907	0.2969	0.3361	0.1550	0.2219	0.9228
Zhejiang	0.6845	0.4491	0.3797	0.3277	0.0677	0.1897	0.2968	1.0000
Chongqing	0.5682	0.3829	0.2671	0.3696	0.1048	0.2070	0.2998	0.7992

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
