# Peer review of "Comparing Economics, Environmental Pollution and Health Efficiency in China"

_ijerph, 2019, doi:10.3390/ijerph16234827_

Round 1

Reviewer 1 Report

Comments to the Author In this paper of “Comparing Economics, environmental pollution and health efficiency in china” The author uses a two-stage dynamic DEA model of considering undesirable outputs) to calculate the total efficiency, production efficiency and health efficiency in 2014-2017. The results show that the total efficiency values of most provinces are between 0.2 and 0.4, which have a large room for improvement; Production efficiency and health efficiency have increased in recent years, but the health efficiency values of most provinces are still so low that dragged back the overall efficiency. I think the topic of this paper has certain advance and important practical significance. Generally speaking, the methods used in this paper are reasonable. The title, abstract, presentation, style, methodology, data, and interpretation of this paper is appropriate Some suggestions: 1. The input-output description of the model is not accurate, which needs further elaboration. Please more in-depth analysis input and output variables. 2. Please add more literature reviews to this paper. I would like to see more discussions of the literature so that I can clearly identify the article relating to the author’s ideas and study. 3. The sources of empirical data are not clear and need to be added 4. Figure 2 needs a more detailed explanation Overall, this paper, according to its content and presentation, need minor revisions.

Author Response

Response to Reviewer 1 Comments

Thank you very much for your helpful feedback and insightful comments. We have taken all of the suggestions and comments into consideration during this revision. We truly appreciate the opportunity to revise our paper, and believe that our manuscript has significantly improved in response to the ideas and recommendations of the Review Team.

Point 1: The input-output description of the model is not accurate, which needs further elaboration. Please more in-depth analysis input and output variables.

Response 1: Thank you for your valuable suggestions. The input-output description of the model has been added in Line431-446.

Set  and  to represent the input and output in each division , with representing divisions to andbeing the  and  division set. The input and output, links, and carryover definitions are given in the following.

Inputs and outputs: refers to input at time period for  division  .

refers to output r in time period t for  division ; if part of the output is not ideal, it is considered an input for the division.

Links:  refers to the period t links from  division  to division , with being the number of  to  links.

Ztj(kh)t ∈R+(j =1;…; n; l = 1;…; Lkh; t = 1;…; T)

Carryovers:

refers to the carryover of  to the  period from  division  to division , with being the number of carryover items in division .

 is the number of input links for each division k,  is the number of output links for each division k, indicates the number of desirable carryovers for each division k, and indicates the number of undesirable carryovers for each division k.

Point 2: Please add more literature reviews to this paper. I would like to see more discussions of the literature so that I can clearly identify the article relating to the author’s ideas and study. 

Response 2: More literature reviews have been added to the paper.

Ministry of Water Resources of the People's Republic of China.China Water Resources Bulletin 2018. National Bureau of Statistics. International Statistical Yearbook. Li, Y.; Chiu, Y.H.; Wang, L.H.; Liu, Y.C.; Chiu,C.R. A Comparative Study of Different Energy Efficiency of OECD and Non-OECD Countries.Tropical conservation science.2019,12. Harrison,R. M . Pollution: causes, effects and control.Journal of Applied Ecology,1996, 21(2):729. Lu,Y.; Song,S.;Wang,R. Impacts of soil and water pollution on food safety and health risks in China. Environment International,2015, 77:5-15. Remoundou,K.; Koundouri,P.Environmental effects on public health: An economic perspective. International journal of environmental research and public health,2016(8):2160-2178. Zhang,X.H.;Deng,S.H.;Wu,J. A sustainability analysis of a municipal sewage treatment ecosystem based on energy. Ecological Engineering, 2010, 36(5):685-696. Pouriyeh,A;Khorasani,N.;Lotfi,F.H. Efficiency evaluation of urban development in Yazd City, Central Iran using data envelopment analysis.Environmental monitoring and assessment.2016,118(11). Yan,D.; Lei,Y.L.; Shi,Y.K.; Zhu,Q.; Li,L.; Zhang,Z. Evolution of the spatiotemporal pattern of PM2.5 concentrations in China - A case study from the Beijing-Tianjin-Hebei region.Atmospheric environment.2018,183:225-233. Rezaee,M.J.; Moini,A.; Asgari,F.H.A. Unified Performance Evaluation of Health Centers with Integrated Model of Data Envelopment Analysis and Bargaining Game.Journal of medical systems.2012,36(6):3805-3815. Yang,W.X.; Li,L.G. Efficiency evaluation of industrial waste gas control in China: A study based on data envelopment analysis (DEA) model.Journal of cleaner production.2018,179:1-11. Xu,X.L.; Zhou,L.L.; Antwi,H.A.; Chen,X. Evaluation of health resource utilization efficiency in community health centers of Jiangsu Province, China.Human resources for health.2018,16.

Point 3: The sources of empirical data are not clear and need to be added.

Response 3: Thank you for your valuable suggestions. The sources of empirical data have been added.

See Line 512-518:The data of Energy consumption, CO2 (calculated by raw coal, coke, crude oil, gasoline, kerosene, diesel, fuel oil, natural gas, electricity) are extracted from China energy statistics yearbook (2014-2017); the data of Employment population, GDP, fixed assets investment are from China Statistical Yearbook (2014-2017); the data of water consumption, COD and chromium emissions are from China Water resources bulletin (2014-2017) while the data of financial health expenditure, the number of health technicians, health index (calculated by hospital admission rates) and population mortality rate are from China health statistics yearbook (2014-2017).

Point 4: Figure 2 needs a more detailed explanation.

Response 4: Figure 2 has been analyzed in detail in the original text. Geographically, the total efficiency values of 30 provinces in the central, western, and eastern analyses were analyzed. According to the trend line in the figure, the trend of the total efficiency value of each province or city in 2014-2017 is analyzed in detail. Line676-683: In the middle area, it can be seen that from 2014 to 2015, the Heilongjiang province and the Nei Monggol have significant decreases, about 0.8, which may be related to the special situation of the year, and then the total efficiency value of Heilongjiang is more gradual, the efficiency of Nei Monggol then rises to 0.6 in 2016 and down to 0.4 again in 2017. Jilin province’s total efficiency value is flat from 2014 to 2015, then it increases to around 0.5 in 2016 and down to around 0.3 in 2017.Shanxi province rose by about 0.8 to 1 from 2014 to 2015, and then maintained a flat trend. In other provinces in the central region, the total efficiency value is between 0.2 and 0.25, which has a large room for improvement.

Reviewer 2 Report

Congratulations for you work. But, the authors have not synthesis capacity

You should improve punctuation usage.

The abstract should have 200 words.

Refrences should be with numeration in the text.

The article should be with format of this journal (Tables, figures, equations, and references)

For line 12-13 “Yet have not considered the use of water resources and the impact of wastewater pollutant emissions on the economy and health”, but you migth considerer:

Harrison, R. M. (2001). Pollution: causes, effects and control (No. Ed. 4). Royal Society of Chemistry.

Lu, Y., Song, S., Wang, R., Liu, Z., Meng, J., Sweetman, A. J., ... & Wang, T. (2015). Impacts of soil and water pollution on food safety and health risks in China. Environment international, 77, 5-15.

Remoundou, K., & Koundouri, P. (2009). Environmental effects on public health: An economic perspective. International journal of environmental research and public health, 6(8), 2160-2178.

Zhang, X., Deng, S., Wu, J., & Jiang, W. (2010). A sustainability analysis of a municipal sewage treatment ecosystem based on emergy. Ecological engineering, 36(5), 685-696.

For line 30-763: Add references

For line 393 – 422: The equations should be with resolution better and description of each variable.

Figures should be improve (urgent)

Conclusion should be very more precises. You don´t separe the ideas (subchapter).

Author Response

Response to Reviewer 2 Comments

Thank you very much for your helpful feedback and insightful comments. We have taken all of the suggestions and comments into consideration during this revision. We truly appreciate the opportunity to revise our paper, and believe that our manuscript has significantly improved in response to the ideas and recommendations of the Review Team. Please see our detailed response attached.

Reviewer 3 Report

This paper consists of a very interesting work of generating production and health efficiency indicators in 30 provinces of China between 2014 and 2017, using two-stage dynamic DEA model.

General Observations
Quoting is missing, in the paragraph between lines 20 to 38, several data were given without citing the source, this happens again in line 40.
On lines 467 and 480 the equations are unnumbered.
Approximate the title of Table 3 to the body of the table.

Methodology
The methodology is well presented which facilitates the understanding and the proposed modeling for the hypothesis analysis.
The analysis of the results is also coherent and clear.

Author Response

Response to Reviewer 3 Comments

Point 1: Quoting is missing, in the paragraph between lines 20 to 38, several data were given without citing the source, this happens again in line 40.

Response 1:  

Thank you for your valuable suggestions. The quoting and data sources have been added between line 20-38.

Since the industrial revolution, while the economy has developed rapidly, water and energy on Earth have also been rapidly consumed. According to statistics, by the end of 2018, China's total water resources reached 279 billion cubic meters, accounting for 6% of the world's total water resources, ranking fourth in the world, but the per capita water resources are only 1/4 of the world average, and the distribution is not evenly.In terms of energy, China's total energy consumption has been in the forefront of the world for several consecutive years [1]. In 2016, it surpassed the United States and became the world's largest energy consumer, accounting for 22.6% of the world's primary energy consumption. In 2018, China's crude oil imports reached a new high, importing 460 million tons crude oil throughout the year, an increase of 10.1% over the previous year [2]. The data shows that China's energy consumption is extremely large, and it is laborious in self-sufficiency.

Point 2: On lines 467 and 480 the equations are unnumbered.

Response 2: The equations have been numbered in the paper.See line 544 and 558.

      (15)   

   (16)                                  

Point 3: Approximate the title of Table 3 to the body of the table.

Response 3: Amend “Table 3. Descriptive Statistics of inputs and outputs.” to Table 3. Descriptive Statistics of variables.

Round 2

Reviewer 2 Report

 Congratulations for your job. You should improve some format details.

This manuscript is a resubmission of an earlier submission. The following is a list of the peer review reports and author responses from that submission.